# Dependency on host vitamin B12 has shaped *Mycobacterium tuberculosis* Complex evolution

Elena Campos-Pardos[1,2], Santiago Uranga [1,2], Ana Picó[1,2], Ana Belén Gómez[1,2] & Jesús Gonzalo-Asensio [1,2] ✉

Human and animal tuberculosis is caused by the *Mycobacterium tuberculosis* Complex (MTBC), which has evolved a genomic decay of cobalamin (vitamin B12) biosynthetic genes. Accordingly, and in sharp contrast to environmental, opportunistic and ancestor mycobacteria; we demonstrate that *M. tuberculosis* (*Mtb*), *M. africanum*, and animal-adapted lineages, lack endogenous production of cobalamin, yet they retain the capacity for exogenous uptake. A B12 anemic model in immunocompromised and immunocompetent mice, demonstrates improved survival, and lower bacteria in organs, in B12 anemic animals infected with *Mtb* relative to non-anemic controls. Conversely, no differences were observed between mice groups infected with *M. canettii*, an ancestor mycobacterium which retains cobalamin biosynthesis. Interrogation of the B12 transcriptome in three MTBC strains defined L-methionine synthesis by *metE* and *metH* genes as a key phenotype. Expression of *metE* is repressed by a cobalamin riboswitch, while MetH requires the cobalamin cofactor. Thus, deletion of *metE* predominantly attenuates *Mtb* in anemic mice; although inactivation of *metH* exclusively causes attenuation in non-anemic controls. Here, we show how sub-physiological levels of B12 in the host antagonizes *Mtb* virulence, and describe a yet unknown mechanism of host-pathogen cross-talk with implications for B12 anemic populations.

Vitamin B12 (B12), historically known as the "anti-pernicious anemia factor" by its ability to cure pernicious anemia, was first documented in 1926[1]. It has the largest and most complex chemical structure of all the vitamins and biological cofactors, whose intricate structure was solved in 1956 by X-ray crystallography[2]. The term B12 -or cobalamin (Cbl)- refers to a group of water soluble, cobalt-containing corrinoid molecules, which are required in a wide range of metabolic processes both in prokaryotes and animals. Specifically, B12 in animals is essential for carbohydrate, fat and protein metabolism, and the formation and regeneration of red blood cells, as well as the maintenance of the central nervous system. Despite most prokaryotes and animals possess enzymes that require B12 as cofactor, only some bacteria and archaea

are able to synthesize B12 de novo[3]. However, even though some bacteria of the microbiota residing in the large intestine produce Cbl, mammals are not able to uptake Cbl produced at this location, since the site of B12 absorption is located in the small intestine. Therefore, animals, including humans, must assimilate B12 by dietary intake. Indeed, some mammalian herbivores (including rabbits, mice, rats and primates) practice coprophagy to obtain Cbl as they fail to obtain sufficient vitamin from vegetarian dietary sources. In humans, to obtain full benefits from vegan and vegetarian diets, these individuals should ingest fortified foods and/or B12 supplements[4].

The presence of B12-dependent enzymes in bacteria and animals makes this vitamin an attractive candidate for host-pathogen cross-

[1]Grupo de Genética de Micobacterias, Departamento de Microbiología. Facultad de Medicina, Universidad de Zaragoza, IIS Aragón, Zaragoza, Spain. [2]CIBER Enfermedades Respiratorias, Instituto de Salud Carlos III, Madrid, Spain. ✉e-mail: jagonzal@unizar.es

talk. Indeed, it has been proposed a role for B12 in shaping the ecological niche of some bacterial genus as *Salmonella* or *Yersinia*. While non-typhoidal *Salmonellae* retain an intact B12 pathway, this is inactivated in the typhoidal human pathogens[5]. Similarly, although *Y. enterocolitic*a and environmental *Yersinia* species retain all the genes for B12 synthesis, the human-pathogens *Y. pseudotuberculosis* and *Y. pestis* have lost most of them[6]. In other words, those serotypes or species maintaining a functional production of endogenous B12 are adapted to cause intestinal infections; while those pathogens lacking B12 biosynthesis are associated with disseminated diseases. Together, maintaining the B12 biosynthesis in enteric pathogens provides a metabolic advantage to outcompete their intestinal microbiota counterparts in an inflamed gut; while loss of B12 synthesis might represent a signature of invasive pathogens for the change of the intestinal niche and a movement away to a more systemic infectious cycle[6].

In this study, we will focus on the role of the host-derived B12 in the virulence of *Mycobacterium*, a genus comprising relevant intracellular pathogens as *M. tuberculosis*, which still causes 1.3 millions of tuberculosis (TB) deaths each year[7]. The *M. tuberculosis* Complex (MTBC) comprises a group of closely related heterotypic synonyms of *M. tuberculosis*, adapted to cause TB in humans and animals[8]. *M. canettii* is considered the progenitor and the putative common ancestor of MTBC members and it is thought that *M. canettii* diverged from the tubercle bacilli before the clonal expansion of the MTBC[9]. The genus *Mycobacterium* also includes non-tuberculous, and environmental, mycobacteria which eventually cause disease in immunocompromised, and chronic obstructive lung disease patients. The existing literature about B12 synthesis in the *Mycobacterium* genus, and more specifically in *M. tuberculosis*, is ambiguous. On the one hand, it has been documented that mycobacteria, and related actinomycetes, generally retain the ability to synthesize B12, predominantly by the aerobic pathway, and the genome sequence of *M. tuberculosis* H37Rv reveals the presence of multiple *cob* genes predicted to function in the de novo biosynthesis of B12[10]. In another study, it was suggested that Cbl biosynthesis pathway is functional and subjected to purifying selection in *M. tuberculosis*[11]. Conversely, on the other hand, our closer inspection of *cob* genes of *M. tuberculosis* and related TB-causing bacteria reveals deletions, insertions, and polymorphisms which may ablate B12 biosynthesis. Here, we aim to disentangle this controversy, and to decipher the role of B12 in the host-pathogen cross-talk of the MTBC.

## Results

### Tubercle bacilli depend on the uptake, not synthesis, of vitamin B12

In spite of being considered highly clonal, the MTBC contains genetic polymorphisms which hypothetically have contributed to shape its evolution and host preference. Specifically, of the 16 *cob* genes putatively involved in Cbl synthesis in *Mycobacterium*[12], we found non-synonymous mutations, insertions and deletions in 10 *cob* genes, which affect different branches of the MTBC (Fig. 1a). We confirmed genomic data by Sanger sequencing of specific clinical isolates belonging to the different MTBC lineages affected by *cob* mutations (Fig. 1a and S1). Of these polymorphisms, it is remarkable the presence of *cobF* and RD9 genomic deletions since they are large polymorphisms affecting several arms of the MTBC. The *cobF* gene is absent in all MTBC lineages, except lineage 8[13]; and consequently, the most widespread lineages of *M. tuberculosis, M. africanum* lineages 5 and 6, and animal-adapted lineages, carry this deletion. Similarly, RD9, which eliminates the 5′-terminus of the *cobL* gene, is absent in *M. africanum* and animal-adapted mycobacteria (Fig. 1a).

Overall, these data indicate that the MTBC have evolved a genomic decadence in B12 synthesis genes. However, being aware that non-synonymous mutations do not necessarily abrogate enzyme activity, and that gene deletions could be compensated by accessory genes, we

assayed B12 production in representative *Mycobacterium* strains. None of the MTBC strains tested, (*M. tuberculosis* strains from lineages 2 and 4; *M. africanum* from lineages 5 and 6; and the animal-adapted *M. bovis*), produced detectable levels of B12 (Fig. 1b). In contrast, the MTBC progenitor, *M. canettii* showed an optimal B12 endogenous synthesis (Fig. 1b). Further, we also demonstrate that environmental (*M. smegmatis*), and opportunistic (*M. avium, M. abscessus, M. fortuitum, M. gordonae, M. mucogenicum, M. xenopii*) mycobacteria produce variable B12 levels (Fig. 1b). This latter result is in agreement with a recent study demonstrating B12 production in various non-tuberculous mycobacteria, but not in *M. tuberculosis*[14].

Once demonstrated that B12 synthesis is abrogated in the MTBC, we reasoned that the presence of B12-dependent enzymes in these pathogens[12] could imply the existence of a B12 uptake mechanism. Therefore, upon incubation of *M. tuberculosis*, *M. africanum* and *M. bovis* with exogenous B12, we confirmed the capacity of these bacteria to internalize the molecule (Fig. 1c). All MTBC species tested were able to uptake two B12 isoforms, namely cyano-Cbl and adenosyl-Cbl, in either exponential (Fig. 1c), and stationary growth phases (Fig. S2). The evolutionary preservation of a mechanism for B12 transport, irrespective of a decayed B12 biosynthetic pathway, emphasizes the relevance to scavenge this molecule for the physiology of the MTBC.

### M. tuberculosis exhibits reduced virulence in B12 anemic mouse models

Our observation that MTBC members have retained the ability to scavenge exogenous B12, despite lacking endogenous synthesis, led us to investigate whether B12 could represent a host factor which modulate *M. tuberculosis* virulence. To prove this hypothesis, we first established B12 anemic mouse models based on feeding rodents during 8 weeks with a B12-deficient diet[15]. We confirmed that immunocompetent C57BL/6, and immunocompromised SCID mice fed without B12 showed 6.5- and 3.4-fold less B12 serum levels, respectively, than their counterparts fed with a normal diet (Fig. 1d). Rodents fed with both diets showed signs of animal welfare, which is in agreement with previous findings that a moderate depletion in B12 serum levels does not impact on the physiological status of mice[16]. Then, we infected B12-defective, and control mice, by the intranasal route with *M. tuberculosis* H37Rv and we evaluated either SCID mice survival times, or bacterial loads in C57BL/6 organs (Fig. 1e). We observed that anemic SCID mice survived significantly better after the *M. tuberculosis* infection than the control group (Fig. 1f and S6). This denotes that, in the absence of adaptive immunity, the virulence of *M. tuberculosis* is reduced under suboptimal B12 serum levels, and *viceversa*.

This observation was confirmed after examination of bacterial loads in lungs and spleen of C57BL/6 immunocompetent animals. To rule out that differences in bacterial loads were due to an in vitro B12-dependent fitness, we first confirmed equivalent *M. tuberculosis* growth rates in laboratory media supplemented with or without B12 (Fig. S3). Depleted B12 serum levels in C57BL/6 mice resulted in significantly lower bacterial replication in the lungs, and lower dissemination to the spleen, relative to the control group (Fig. 1g and S7); reinforcing the hypothesis that *M. tuberculosis* needs optimal host B12 serum levels to develop a complete virulence. Then, we tested a *M. canettii* infection in our mouse anemic models, because unlike *M. tuberculosis*, this ancestor mycobacterium is able to endogenously produce its own B12 (Fig. 1b). First, we optimized the inoculum of *M. canettii* since this bacterium is known to be less persistent and less virulent than *M. tuberculosis*[9]. In order to obtain equivalent survival in SCID mice, it was required from 100–1000-fold higher *M. canettii* inoculum relative to *M. tuberculosis* (Fig. S4). Once established the infection conditions, we found equivalent survival times in SCID mice infected with *M. canettii*, independently of the B12 serum levels of mice (Fig. 1f and S6). Further supporting the independence of *M. canettii* virulence and host B12 status, non-

significant differences in lungs and spleen bacterial loads were observed in C57BL/6 mice fed with normal or B12-depleted diets upon infection with the ancestor *M. canettii* (Fig. 1g and S7). *Post-mortem* analysis of *M. tuberculosis*- and *M. canettii*-infected mice confirmed that B12 serum levels remained low in anemic animals relative to control groups throughout the curse of the experiments (Fig. S5). To confirm the link between *M. canetti* virulence and its intrinsic ability to produce B12, we constructed a *M. canettii* Δ*cobMK* mutant unable to synthesize this molecule (Figs. S8 and S15). This *M. canettii* B12 mutant was used to infect B12-deficient mice and we found that, unlike the wild type, the mutant resulted strongly attenuated as measured by longer SCID survival, and significantly lower bacteria in C57BL/6 organs (Fig. S8). Together, results indicate that host B12 levels directly correlate with *M. tuberculosis* virulence, but this association does not apply to the ancestor of the MTBC due to its specific ability to synthesize B12.

## B12 supplementation of anemic mice recovers M. tuberculosis virulence

Results described above led us to speculate that restoring healthy B12 serum levels in anemic mice might impact on the virulence of *M. tuberculosis* in infected animals. Accordingly, 4 groups of SCID mice with different B12 treatments were established: a standard diet control, 2 groups fed with independent B12-deficient diets, and another group receiving weekly subcutaneous B12 supplementation after being fed with a B12-deficient diet (Fig. 2a). Following each B12 treatment, all animal groups were intranasally infected with *M. tuberculosis* and we interrogated survival times and B12 serum levels at sacrifice. We found that both B12-defective diets resulted in significantly reduced serum B12 levels relative to the control diet. Further, supplementation with B12 resulted in efficient restoration of serum levels of this vitamin, comparable to those of the control mice (Fig. 2b). Survival times of each mice group remarkably correlated to their respective B12 serum

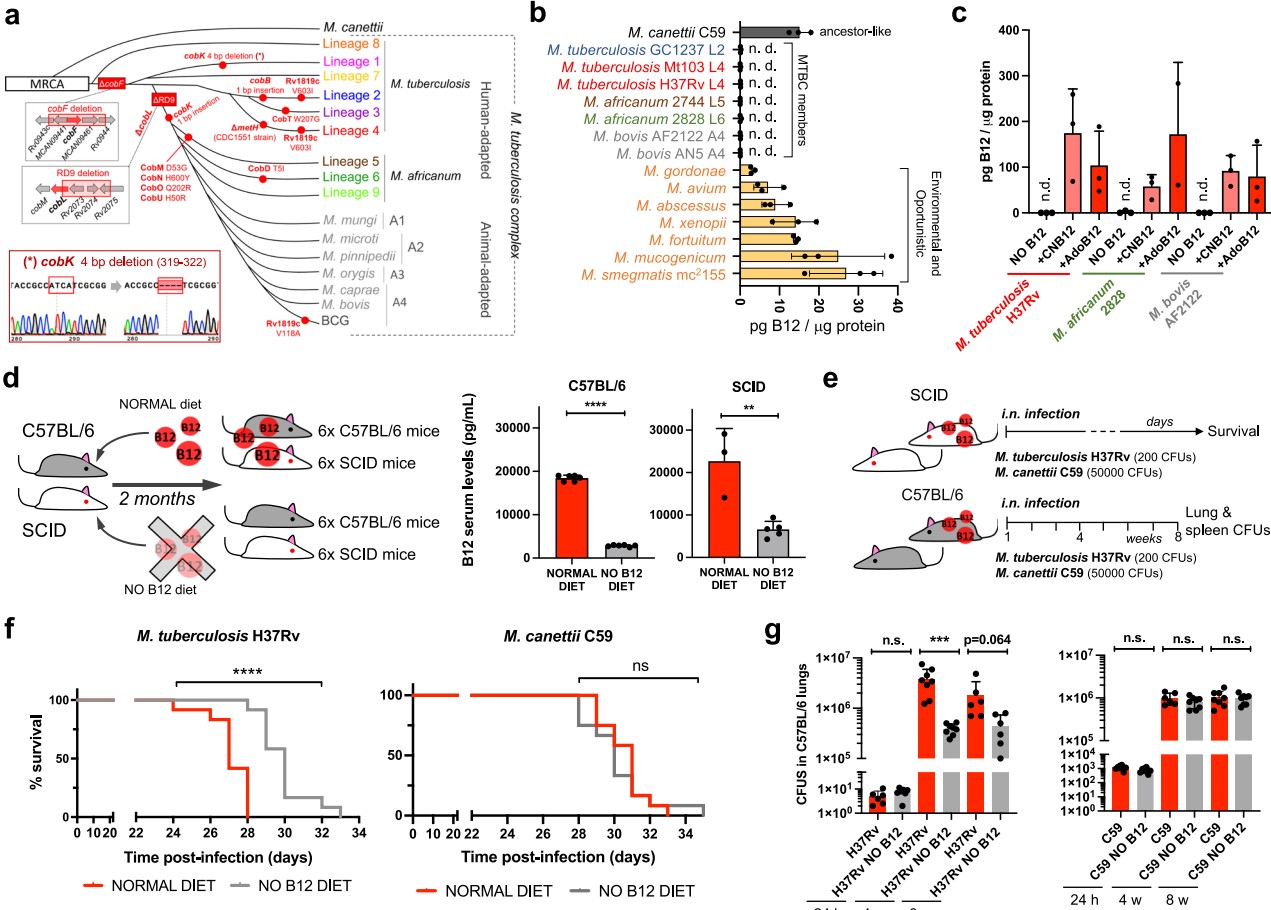

**Fig. 1 | M. tuberculosis, but not its ancestor M. canettii, exhibits reduced virulence in B12 deficient mice models due to the inability of MTBC bacteria to synthesize endogenous B12. a** Identification of mutations (red dots) and genomic deletions (red boxes) affecting the de novo B12 biosynthesis in MTBC strains and in its last known common ancestor (*M. canettii*). The bottom left box shows the validation by Sanger sequencing of one of these mutations. Additional Sanger sequencing confirmations are provided in Fig. S1. **b** B12-levels synthesized by MTBC species (color codes matches those in panel A), in *M. canettii* (dark gray) and in environmental and opportunistic mycobacteria (yellow). "n.d." denotes non detected. Bars and error bars are the average and SD from three biological replicates. **c** Cyanocobalamin (CNB12) and adenosylcobalamin (AdoB12) scavenged by MTBC members in cultures containing B12 grown until exponential growth-phase. Data are the mean and standard deviation from three biological replicates. **d** Development of mouse B12 anemic models and confirmation of decreased serum

B12-levels in the treated groups. Graph data are mean ± SD of *n* = 6 and *n* ≥ 3 biological replicates of C57BL/6 and SCID mice, respectively. Statistical analysis was performed using unpaired *t*-test. **e** Experimental set-up of the survival and virulence experiments in SCID and C57BL/6 anemic mice infected with the wild type strains of *M. tuberculosis* H37Rv and *M. canettii* C59. **f** Survival rates from SCID mice fed with each diet and inoculated by the intranasal route with *M. tuberculosis* H37Rv or *M. canettii* C59. Each curve represents the pool from 2 independent experiments (*n* = 12). Statistical analysis was performed using Log-rank (Mantel-Cox) test. **g** Bacterial loads in the lungs of control and anemic C57BL/6 mice after 24 h, 4 weeks, or 8 weeks infected with *M. tuberculosis* H37Rv or *M. canettii* C59. Data represent mean ± SD of *n* ≥ 6 biological replicates per mice group. Statistical analysis was performed using unpaired *t*-test. *p*-values are as follows: ****0.0001 > *p*; ***0.001 > *p* > 0.0001; **0.01 > *p* > 0.001; ns not significant, *p* ≥ 0.05.

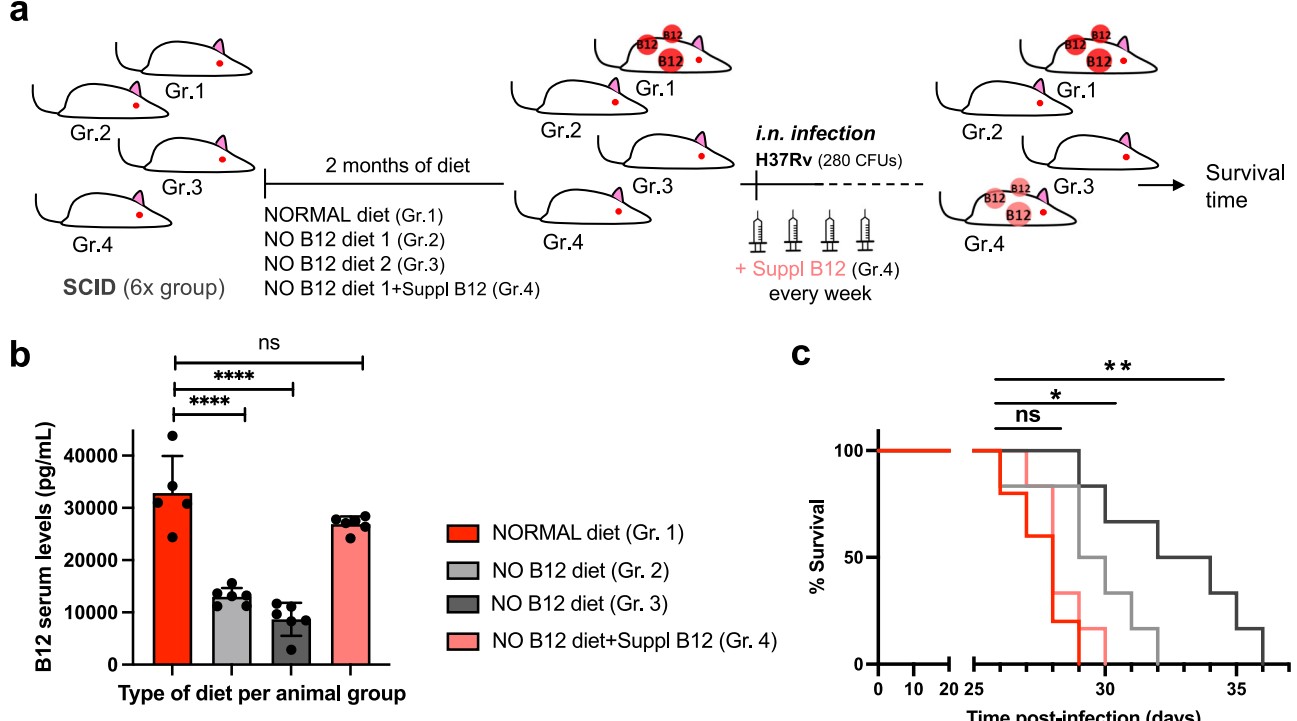

**Fig. 2 | Subcutaneous B12 supplementation efficiently restores B12 serum levels in anemic mice and results in enhanced M. tuberculosis virulence.**
**a** Experimental design and timelines of the different B12 treatments, B12 subcutaneous supplementation, and *M. tuberculosis* infection, in each SCID mice group. **b** B12 serum levels in animal groups described in panel (**a**) after sacrifice at the humane endpoint. B12 levels remained significantly decreased in animals treated with B12 restricted diets relative to the control group throughout the curse of the experiment. Note that weekly B12 supplementation restored standard B12 serum levels in mice. Graph data are mean ± SD of $n = 6$ biological replicates. Statistical analysis was performed using One-Way ANOVA and Tukey post-test. **c** Survival rates from groups of 6 SCID mice for each type of treatment after infection by the intranasal route with *M. tuberculosis*. Statistical analysis was performed using Log-rank (Mantel-Cox) test. *p*-values are as follows: ****$0.0001 > p$; **$0.01 > p > 0.001$; *$0.05 > p > 0.01$; *ns* not significant, $p \geq 0.05$.

levels. Interestingly, the gradual decrease in B12 serum levels observed after treatment with the different diets translated into longer survival times and consequently into reduced *M. tuberculosis* virulence (Fig. 2b, c). Confirming this observation, restoration of healthy B12 levels in anemic mice by subcutaneous supplementation resulted in a *M. tuberculosis* virulence indistinguishable from the mice group fed with a standard B12-containing diet (Fig. 2c). Together, this confirms that sub-physiological levels of B12 in mice causes *M. tuberculosis* attenuation, and highlights the importance to assimilate B12 from the host for this pathogen.

## The B12-dependent transcriptome in Mycobacterium establishes a link with methionine metabolism

Having demonstrated that MTBC bacteria scavenge B12 from the host to modulate virulence, we sought to identify the molecular mechanism(s) responsible for this phenotype. We studied bacteria from the most representative lineages causing TB in humans and animals. Specifically, *M. tuberculosis* GC1237 and H37Rv were selected as representative of human-adapted strains from lineages 2 (sublineage Beijing) and 4, respectively; and *M. bovis* (strain AF2122) was selected as a representative animal-adapted strain. Replicates from each strain were grown in the presence, or absence, of exogenous B12, bacterial RNA was sequenced by RNA-seq, and differential gene expression between both conditions was interrogated (Supplementary Data 1–3). This analysis produced three differential expression sets, one per each MTBC strain analyzed, which were merged to gain insight into the B12-dependent transcriptome in the MTBC (Fig. 3a). We identified 7 genes organized into 2 genome clusters, which showed a robust down-regulation in the presence of B12 (Fig. 3b and S9). All these genes exhibited B12-dependent repression across replicates in all

*Mycobacterium* strains tested, which probes that the identified B12-dependent transcriptome is reproducible and extensive to the MTBC (Fig. 3a, b, S9 and Supplementary Data 1–3). To additionally confirm the B12 regulation of the identified genes, we corroborated down-regulated expression of *Rv1129c*, *prpD*, *metE*, *PPE2*, and *cobQ1* measured by quantitative-PCR in *M. tuberculosis* H37Rv cultures incubated with B12 relative to controls (Fig. 3c and S9). Further, we conducted a Multiple Reaction Monitoring (MRM-MS) proteomic approach to measure PrpD, PrpC and MetE protein levels in *M. tuberculosis* GC1237 and H37Rv cultures treated with or without B12. Results demonstrated a complete absence of PrpD and PrpC, and an average 9.8-fold reduction of MetE protein, when B12 is added to the cultures (Fig. 3d and S10). In a previous independent study with the H37Rv strain, the authors also identified PrpR, PrpDC, Rv1132 and MetE as B12-regulated genes, providing robustness and cumulative knowledge about the B12 regulatory network in *M. tuberculosis*[17].

We then focused on the genome cluster containing 5 out of 7 genes of the B12-dependent transcriptome Rv1129c (PrpR), the *prpC-prpD-Rv1132* operon, and the *metE* gene (Fig. 3B). Provided that the role of PrpR, and PrpDC, has been studied in detail[18–22] as will be discussed below, we decided to focus our study in the B12-regulated *metE* gene, whose implications in the virulence of *M. tuberculosis* are less understood.

## A M. tuberculosis metE mutant is predominantly attenuated in anemic mice

The final step of methionine synthesis in *M. tuberculosis* is catalyzed by two complementary methionine synthases: MetE or MetH. The activity of both proteins is regulated by B12, albeit by different mechanisms. Expression of *metE* is regulated by a B12 riboswitch

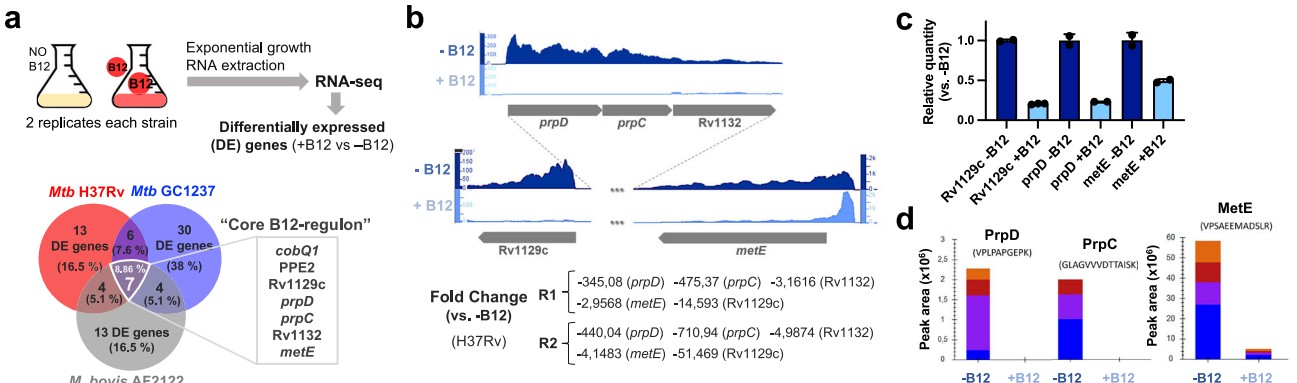

**Fig. 3 | The core B12-dependent transcriptome identifies a gene cluster regulated by B12 in the MTBC and establishes a link with methionine metabolism.** **a** Experimental procedure for RNA-seq analysis of differentially expressed (DE) genes in response to B12 supplementation of MTBC cultures. Venn diagrams show the resulting DE genes from two independent replicates of *M. tuberculosis* H37Rv (L4), *M. tuberculosis* GC1237 (L2), and the animal-adapted *M. bovis* AF2122. Those DE genes common to the three strains are indicated as the "core B12-regulon", and consist on 7 downregulated genes in the presence of B12. **b** RNA-seq profiles showing downregulated gene expression of the gene cluster Rv1129c (*prpR*)-*prpD*-*prpC*-Rv1132-*metE* in *M. tuberculosis* H37Rv in response to exogenous B12 supplementation (light blue) relative to the experimental control without B12 (dark blue). Note that in order to visualize expression differences, each genetic section has been represented with its corresponding scale. Expression profiles of the remaining B12-dependent genes are indicated in Fig. S9. **c** qRT-PCR measures of Rv1129c (*prpR*), *prpD* and *metE* in *M. tuberculosis* H37Rv cultures grown with or without B12. Relative quantity refers to the differential expression of the selected genes in the presence of B12 in comparison with its expression in the absence of B12. Each gene was normalized against *sigA* expression in each sample. Graphs represents mean ± SD from three biological replicates. Expression of additional B12-dependent genes is shown in Fig S9. **d** Quantification of PrpD, PrpC and MetE protein levels of *M. tuberculosis* H37Rv both in presence and absence of B12 by MRM-MS. Bars represent the area under the curve for every transition from a specific peptide of each protein in both experimental conditions. Equivalent results were obtained for the rest of the analyzed peptides (Fig. S10) in two biological replicates of H37Rv and GC1237 strains of *M. tuberculosis*.

located upstream the *metE* coding sequence, while MetH requires B12 as cofactor[23]. Our transcriptomic and proteomic results demonstrate decreased *metE* mRNA and protein expression when B12 is present (Fig. 3b–d), which is in agreement with the regulatory mechanism exerted by the riboswitch, and with previous observations[23]. We constructed a Δ*metE* mutant in *M. tuberculosis* H37Rv (Fig. S15) to understand the role of this gene in virulence modulation by B12. This Δ*metE* mutant is still able to synthesize methionine by the B12-dependent MetH, and consequently the Δ*metE* mutant is predicted to strongly depend on B12 supplementation (Fig. 4a). We confirmed a total absence of in vitro growth, in liquid and solid media, of the Δ*metE* mutant relative to *M. tuberculosis* H37Rv wild type in the absence of B12 (Fig. 4b, c). Raising B12 concentrations to 0.01 μg/mL partially recoved growth of the Δ*metE* mutant, and 0.1 μg/mL resulted in a complete growth rescue on agar plates (Fig. S11). In addition, we wondered whether *M. tuberculosis* might scavenge L-methionine from the environment. Cultivation of the *M. tuberculosis* Δ*metE* mutant in L-methionine-containing plates without B12 neither resulted in appreciable growth (Fig. 4c), indicative of lack of exogenous L-methionine foraging. To further confirm these phenotypes, we also generated Δ*metE* mutants in wild type *M. canettii* and in a *M. smegmatis* strain defective in endogenous B12 synthesis by deletion of the *cobLMK* operon (Fig. S15). Both mutants exhibited the strong dependency on exogenous B12 to restore in vitro growth to wild type levels previously observed in *M. tuberculosis* (Fig. S12), indicative that methionine synthesis by MetH is evolutionary conserved in *Mycobacterium*. Surprisingly, the growth defect in the absence of B12 showed by Δ*metE* mutants of *M. canettii* and B12-deficient *M. smegmatis*, was successfully rescued in the presence of exogenous L-methionine on agar plates (Fig. S12). This indicates that ancestor and environmental mycobacteria have retained a mechanism for L-methionine transport when growing on solid media, which is lost in *M. tuberculosis*. However, a previous study demonstrated the ability of a *M. tuberculosis* Δ*metA* auxotroph to recover growth in liquid media after supplementation with L-methionine[24]. Based on

this observation, we also confirmed the ability of the Δ*metE* mutant to recover planktonic growth upon supplementation with L-methionine (Fig. S13). Overall, this dependency on exogenous L-methionine of the Δ*metE* mutant when growing on solid, but not in liquid media, is intriguing. We can hypothesize that either L-methionine bioavailability is higher in liquid media, facilitating its assimilation by *M. tuberculosis*; or that *M. tuberculosis* expresses L-methionine transport mechanisms exclusively under planktonic growth. Thus, to gain insight into the virulence implications of *metE* inactivation, and to evaluate whether *M. tuberculosis* can directly access host B12, we evaluated the in vivo phenotype of this mutant under a physiological scenario.

We found a significant attenuation of the mutant measured by improved survival times (Fig. 4d), and lower bacterial loads (Fig. 4e and S7), in SCID and C57BL/6 mice fed with a conventional diet, respectively. This attenuation was much more noticeable upon infection of anemic mice, according to the B12 requirements of the *M. tuberculosis* Δ*metE* strain (Fig. 4f). Comparatively, median survival times of the mutant were 30 days longer than the wild type in SCID anemic mice, compared to 8 days in non-anemic mice fed with routine diet. With regards to lung bacterial loads in C57BL/6 mice, the mutant showed lower bacterial burden when infecting B12-deficient animals (Fig. 4g and S7). These results support the *M. tuberculosis* dependency on host B12 for L-methionine synthesis mediated by MetH. In addition, the remnant attenuation of the *M. tuberculosis* Δ*metE* mutant in mice fed with a conventional B12 diet (Fig. 4d, e), might indicate that in vivo L-methionine synthesis by MetH is far from optimal, due to either insufficient B12 scavenging from the host, or to inadequate enzymatic activity of MetH. Additionally, the overt attenuation of the *M. tuberculosis* Δ*metE* strain in vivo resembles its growth defects on solid media. Having demonstrated the lack of L-methionine transport on agar plates (Fig. 4c), it is tempting to speculate that either *M. tuberculosis* exhibits suboptimal L-methionine uptake from the host, or that L-methionine intracellular concentration is too low to support optimal growth of *M. tuberculosis*.

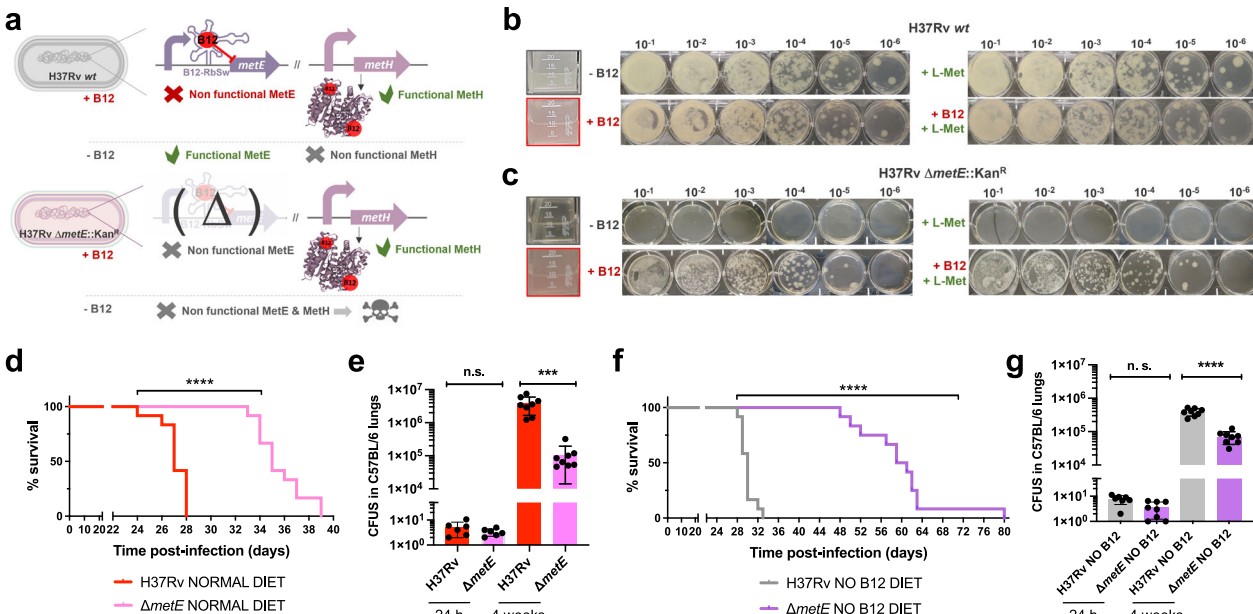

**Fig. 4 | Inactivation of the methionine synthase MetE impacts M. tuberculosis growth in vitro and virulence in vivo under B12 deficient conditions. a** Expected phenotypes of the *M. tuberculosis* H37Rv Δ*metE* knockout compared to its wild type strain in presence and absence of B12. **b** and **c** in vitro growth in liquid (boxed images) and in solid media of the *M. tuberculosis* H37Rv wild type strain (**b**) and its Δ*metE* mutant (**c**) in the absence or presence of exogenous B12 and/or L-methionine. **d** and **f** Survival rates from SCID mice inoculated by the intranasal route with *M. tuberculosis* H37Rv and H37Rv Δ*metE* fed with normal diet (**d**) and a B12-deficient diet (**f**). Each curve represents the pool from 2 independent experiments ($n = 12$). Statistical analysis was performed using Log-rank (Mantel-Cox) test. **e** and **g** Bacterial loads in the lungs of normal (**e**) and anemic (**g**) C57BL/6 mice after 24 h, or 4 weeks infected with *M. tuberculosis* H37Rv and its Δ*metE* mutant. Data are mean ± SD of $n \geq 6$ biological replicates per mice group. Statistical analysis was performed using unpaired *t*-test. *p* values are as follows: ****$0.0001 > p$; ***$0.001 > p > 0.0001$; $p \geq 0.05$; ns: not significant. Results show that inactivation of the methionine synthase MetE attenuates the *M. tuberculosis* virulence more markedly in B12 anemic mice relative to controls.

## Suppressive mutations in the B12-dependent metE riboswitch relieve the attenuation of a M. tuberculosis metH mutant in non-anemic mice

For a complete understanding of the implications of the B12-dependent methionine metabolism in *M. tuberculosis*, we also studied the phenotype of a *M. tuberculosis* Δ*metH* mutant (Fig. S15). This mutant still maintains L-methionine production by MetE, and accordingly, it is expected to have growth defects in the presence of B12 due to the inhibition of the *metE* riboswitch (Fig. 5a)[23]. Once constructed, the *M. tuberculosis* Δ*metH* mutant was inoculated in laboratory media with or without B12 confirming absent, or reduced growth, in liquid and solid media, respectively, when B12 is present (Fig. 5b, c). Next, we confirmed that concentrations of 0.01 µg/mL B12 were sufficient to partially inhibit the growth of *M. tuberculosis* Δ*metH* on solid plates, otherwise indicative of inhibition of the *metE* riboswitch (Fig. S11). We also used this mutant to confirm the absence of exogenous L-methionine uptake in *M. tuberculosis* on agar plates, since supplementation of B12 plates with L-methionine did not restore bacterial growth to wild type levels (Fig. 5b, c). In contrast, a Δ*metH* mutant constructed in *M. canettii* (Fig. S15), grew equivalently to its parent strain in solid media supplemented with B12 and L-methionine (Fig. S12), confirming that L-methionine uptake from solid media is evolutionarily maintained in ancestor mycobacteria, but not in *M. tuberculosis*. As previously demonstrated with the Δ*metE* mutant, we confirmed that assimilation of exogenous L-methionine in the Δ*metH* mutant occurs under planktonic growth, but not during growth on agar plates (Fig. S13). In vivo evaluation of the *M. tuberculosis* Δ*metH* mutant showed reduced virulence in SCID mice (Fig. 5d and S7), and reduced lung bacterial loads in C57BL/6 mice (Fig. 5e) under a conventional diet. In contrast, no differences in bacterial virulence were observed when animals were foraged with a B12 deficient diet (Figs. 5f, g and S7), indicating that suboptimal B12 serum levels are insufficient

to repress the *metE* riboswitch in vivo. We also sequenced the *metE* riboswitch region from 22 *M. tuberculosis* Δ*metH* colonies grown on solid media supplemented with B12, and 30 colonies from lungs of mice infected with this strain and fed with normal diet. As expected, we found that 17/22 of the colonies grown on solid media contained mutations in the riboswitch (Fig. 5h and S14). Surprisingly, only a single colony from the mouse lungs contained mutations in this region, which might indicate that physiological levels of B12 do not impose a selective pressure as high as that observed in vitro. Mapping of the mutations to the predicted structure of the riboswitch demonstrated that these polymorphisms were regularly distributed, and different polymorphisms affecting invariant residues of B12 riboswitches[25] arose independently in independent colonies (Fig. 5i). This result suggests that suppressor mutations in the B12 riboswitch could alleviate the B12 repression of the *metE* gene, and favor the appearance of *M. tuberculosis* Δ*metH* escape mutants when B12 is present, preferentially during growth in vitro.

This finding led us to hypothesize that introduction of a B12-independent *metE* gene into the *M. tuberculosis* Δ*metH* mutant should rescue the B12 growth defects of this strain without the need to evolve suppressor mutations in the *metE* riboswitch (Fig. 6a). We placed the Ag85A promoter (Pr-Ag85A) immediately upstream of the *metE* coding region, and this construction was introduced in the chromosome of the *M. tuberculosis* Δ*metH* (Fig. 6a and S15). Improved expression of the Pr-Ag85A-controlled copy of *metE* was confirmed with respect to *M. tuberculosis* wild type and its *metH* mutant (Fig. 6b). Then, we confirmed that the Pr-Ag85A-*metE* copy enabled the *M. tuberculosis* Δ*metH* to grow in liquid and solid media containing B12, in contrast to the parent *M. tuberculosis* Δ*metH* strain (Fig. 6c). The Pr-Ag85A-*metE* complemented strain was tested in vivo in mice fed with a B12 containing diet, to confirm that survival in SCID mice and bacterial loads in C57BL/6 lungs were equivalent to those of the wild type strain

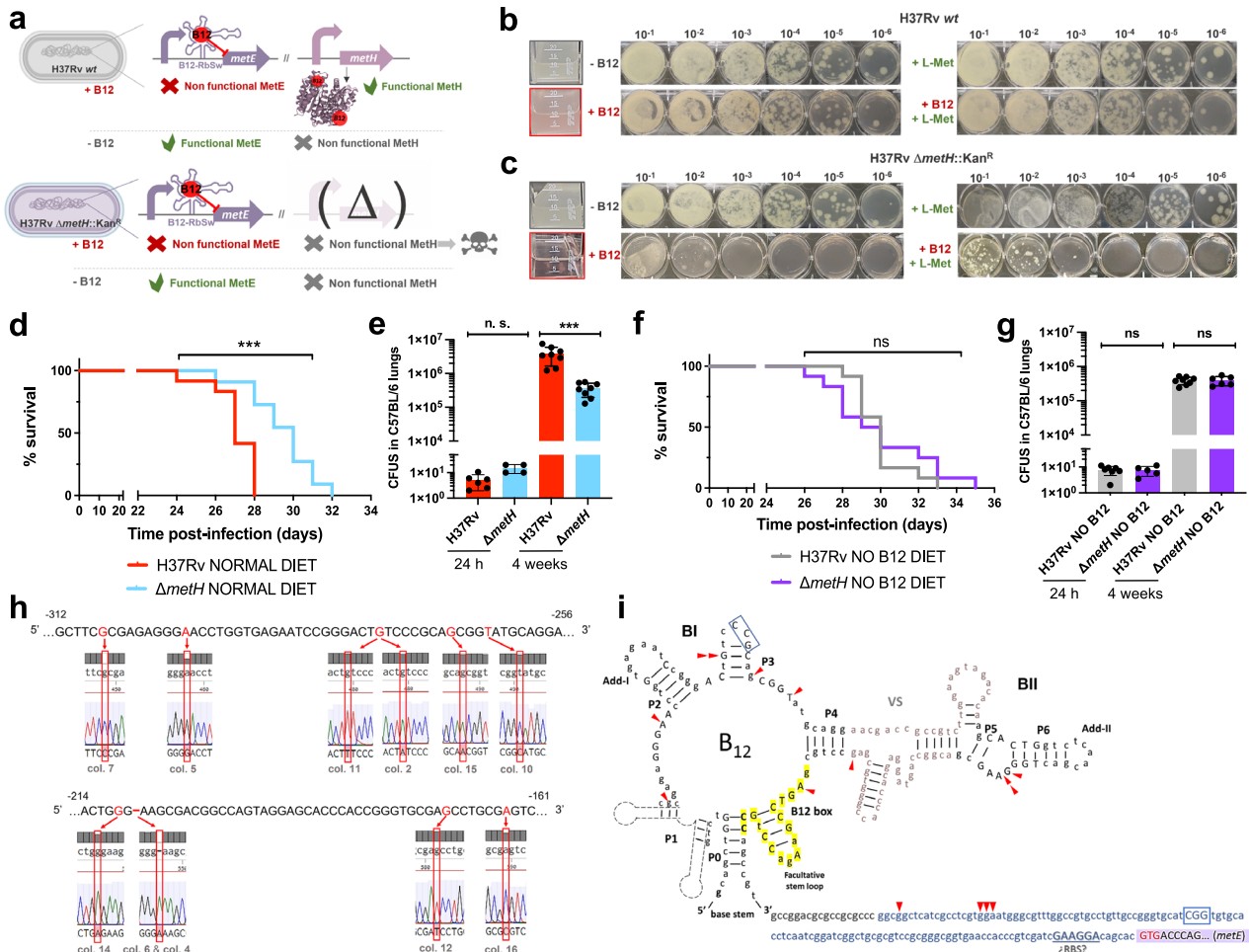

**Fig. 5 | Inactivation of the methionine synthase MetH causes B12 toxicity in M. tuberculosis resulting in alleviating compensatory mutations in the metE B12-Riboswitch. a** Expected phenotypes of the *M. tuberculosis* H37Rv Δ*metH* knockout compared to the wild type strain in presence and absence of B12. **b** and **c** in vitro growth in liquid and in solid media of the *M. tuberculosis* H37Rv wild type strain (**b**) and its Δ*metH* mutant (**c**) in absence or presence of exogenous B12 and/or L-methionine. **d** and **f** Survival rates from groups of SCID mice fed with normal diet (**d**) and B12-deficient diet (**f**), and inoculated by the intranasal route with *M. tuberculosis* H37Rv and the Δ*metH* mutant. Each curve represents the pool from 2 independent experiments (*n* = 12). Statistical analysis was performed using Log-rank (Mantel-Cox) test. **e** and **g** Bacterial loads in the lungs of control (**e**) and B12 anemic (**g**) C57BL/6 mice after 24 h, or 4 weeks infected with *M. tuberculosis* H37Rv and its Δ*metH* mutant. Data are mean ± SD of *n* ≥ 6 biological replicates per mice group. Statistical analysis was performed using unpaired *t*-test. *p*-values are as

follows: ***$0.001 > p > 0.0001$; ns not significant, $p \geq 0.05$. Results show that inactivation of the MetH isoform attenuates the mutant exclusively in animals with normal serum B12-levels (**h**) 5′ UTR sequences of the *metE* B12-Riboswitch. Negative numbers indicate nucleotides immediately upstream of the *metE* initiation codon. Bases showing mutations in *M. tuberculosis* H37Rv Δ*metH* colonies grown with B12 are highlighted in red in the Sanger chromatograms. Aditional mutations are shown in Fig. S14. **i** Secondary structure prediction of the *metE* B12-riboswitch. Location of the identified mutations in the B12-resistant *M. tuberculosis* H37Rv Δ*metH* colonies are indicated by red arrows. Invariant residues across ~200 B12-riboswitches are indicated in capital letters. Lowercase letters indicate *M. tuberculosis* specific residues. The conserved B12-box is highlighted in yellow. The *metE* 5′UTR region is shown in blue letters with the possible RBS underlined. The start of the *metE* CDS is shown with a purple arrow and the initiation codon is highlighted in red.

(Figs. 6d, e and S7). This later result was also reproduced in animals fed with a B12-deficient diet (Fig. 6f, g and S7). Together, our results confirm the role of B12 suppressor mutations in the *metE* riboswitch to relieve the pressure imposed by B12 over the *M. tuberculosis* Δ*metH* strain in vivo.

## Discussion

Humans have evolved through different dietary scenarios which might have shaped not only their nutritional requirements, but their susceptibility to infections. The human evolution has circumvented the need to supply our bodies with dietary B12. Primates reminiscent of our ancestor hominids, show optimal B12 serum levels in their natural habitats obtained from either coprophagy, occasional carnivory, or foraging insects[26]. Present-day tribal societies, resembling our hunters-gatherers pre-Neolithic ancestors, show healthy B12 serum levels probably obtained by sporadic feeding on

hunted meat[27]. Therefore, it is appealing to speculate that during the co-evolution of humans and pathogens, the latter have developed mechanisms to sense B12 in the host environment, and consequently, to promote pathogenesis. Here, we propose the host-derived B12 as such a signaling molecule employed by *M. tuberculosis*. Indeed, the presence of B12 sensing riboswitches has been documented in a plethora of prokaryotes, including intracellular pathogens as *Mycobacterium, Listeria,* or *Salmonella*[28]. These latter bacteria are known to reside within the macrophage phagolysosome sometime in their life cycles. Thus, it might not be casual that B12 is internalized within human cells through the lysosomal pathway[29,30], a cellular compartment expected to establish a close contact with intracellular bacteria. Indeed, two independent studies with *Salmonella*, using either a pernicious anemia mice model, or a B12-depleted diet model, have demonstrated that B12 deficiency renders mice more susceptible to *Salmonella* infection[31,32].

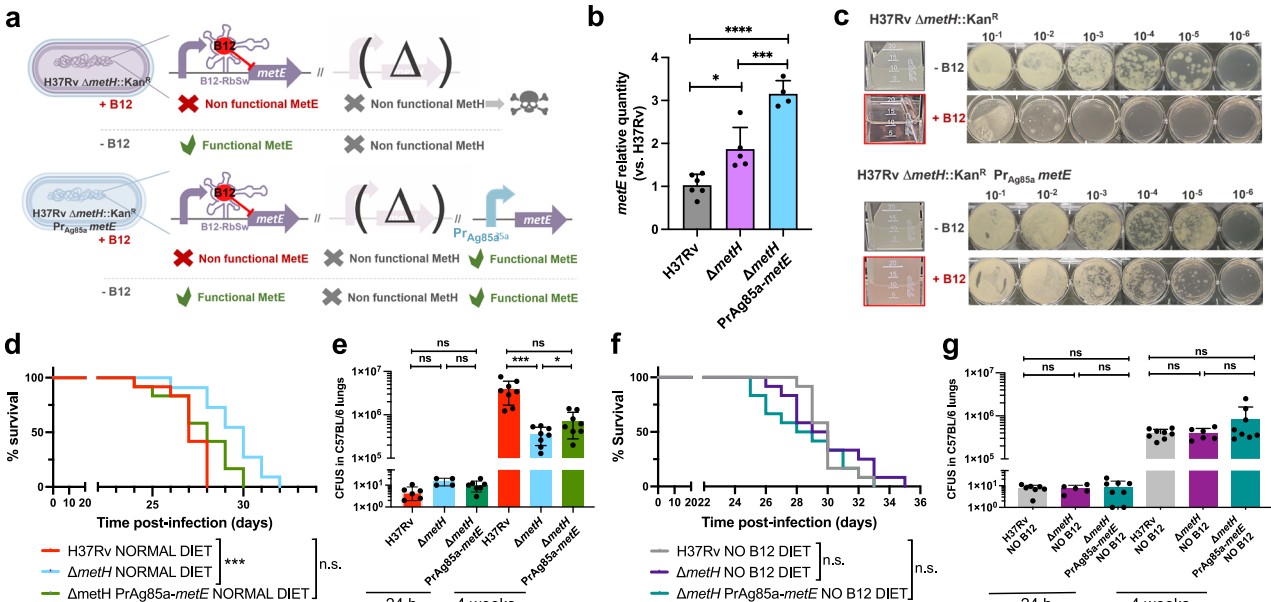

**Fig. 6 | Complementation of the M. tuberculosis ΔmetH knockout with a B12-independent metE gene restores virulence to wild type levels. a** Expected phenotypes of the *M. tuberculosis* H37Rv Δ*metH* Pr<sub>Ag85a</sub>*metE*::Kan complemented strain compared to the parental Δ*metH* strain in presence and absence of B12. **b** Validation by qRT-PCR of the *metE* complementation and its stable expression in the complemented mutant *M. tuberculosis* H37Rv Δ*metH* Pr<sub>Ag85a</sub>*metE*::Kan compared to the Δ*metH* mutant and the wild type strains. Relative quantity refers to the differential *metE* gene expression in each strain in comparison with its expression in the wild type H37Rv. Each gene was normalized against *sigA* expression in each sample. Graphs represents mean ± SD from one experiment with three replicates. Statistical analysis was performed using two-way ANOVA followed by Tukey's multiple comparisons post-test. *p*-values are as follows: ****0.0001 > *p*; ***0.001 > *p* > 0.0001; *0.05 > *p* > 0.01. **c** Experimental growth in liquid (boxed images) and in solid media of the *M. tuberculosis* H37Rv Δ*metH* and the Δ*metH*

Pr<sub>Ag85a</sub>*metE*::Kan strains in presence and absence of B12. **d** and **f** Survival rates from groups of SCID mice fed with normal diet (**d**) or B12-deficient diet (**f**), and inoculated by intranasal route with *M. tuberculosis* H37Rv, its Δ*metH* mutant or the Δ*metH* Pr<sub>Ag85a</sub>*metE*::Kan complemented strain. Each curve represents the pool from 2 independent experiments (*n* = 12). Statistical analysis was performed using Log-rank (Mantel-Cox) test. **e** and **g** Bacterial loads in the lungs of normal (**e**) and B12 anemic (**g**) C57BL/6 mice after 24 h, or 4 weeks infected with *M. tuberculosis* H37Rv, the Δ*metH* knockout, or the Δ*metH* Pr<sub>Ag85a</sub>*metE*::Kan Data are mean ± SD of *n* ≥ 6 biological replicates per mice group. Statistical analysis was performed using unpaired *t*-test. *p*-values are as follows: ***0.001 > *p* > 0.0001; *0.05 > *p* > 0.01; ns: not significant, *p* ≥ 0.05. Results show that complementation with a B12-independent *metE* restores virulence in the *M. tuberculosis* Δ*metH* mutant independently of the B12 status of the host.

In *Mycobacterium*, a recent study used a combination of genetically diverse mice and a collection of *M. tuberculosis* mutants, with the objective of identifying bacterial virulence requirements in the context of a heterogeneous host genetics and immunity. Among their findings, *bacA*, *mutB*, and *PPE2* mutants were negatively selected when infecting some mice strains[33]. Of this pathway, *bacA* is a *M. tuberculosis* B12 transporter[34,35], *mutB* encodes one of the constituents of the B12-dependent MutAB methylmalonyl-CoA-mutase[36], and *PPE2* is regulated by a B12 riboswitch[23]. Even if this experimental design does not resemble our current study, this independent finding complementarily supports the importance of B12 during infection in murine models of TB. Nevertheless, it might be argued that our results obtained with two genetically independent, SCID and C57BL/6, laboratory mouse models, could not be extrapolated to all mice strains. Another possible limitation is related to the lack of testing of *M. tuberculosis* Δ*metE*, or Δ*metH*, complemented strains in vivo to rule out unrelated virulence phenotypes caused by spontaneous mutations arisen during the mutant construction. However, some arguments support our findings in vivo: (i) we used two independent mutant clones of each, the Δ*metE* and Δ*metH* strains, in separate mouse assays to reduce the chance of observing effects from unrelated mutations; (ii) the Δ*metE* and Δ*metH* mutants exhibit complementary phenotypes, adding robustness to the study by demonstrating the main hypothesis with two independent *M. tuberculosis* mutants; (iii) the effect of the Δ*metH* mutation is indistinguishable from the wild type in the context of B12 deficiency; and (iv) virulence attenuation of the Δ*metH* mutation is successfully recovered upon introduction of a functional L-methionine synthesis pathway in vivo.

We and other have studied the effect of B12 on the *M. tuberculosis* metabolism[17]. Of the B12-regulated genes, PrpR acts as a transcription factor of PrpD and PrpC[18], which are involved in the methylcitrate cycle required for optimal in vitro growth on propionate as carbon source[19]. In addition, PrpR or PrpDC, are essential for *M. tuberculosis* multiplication in macrophages, which is probably attributable to the inability of the mutants to metabolize propionate derived from host cholesterol[22]. Surprisingly, a *prpDC* mutant failed to produce differences in pathology, organ bacterial loads, or persistence, relative to the wild type during the mouse infection[19]. The absence of *prpDC* phenotypes in vivo is probably attributable to the metabolic rescue of the methylcitrate cycle in the presence of host B12[20–22]. The great versatility of carbon metabolism in *M. tuberculosis* allows this pathogen to utilize two complementary pathways in vivo for propionate detoxification: the B12-independent methylcitrate cycle, and the B12-dependent methylmalonate pathway. Thus, abrogation of the methylcitrate cycle in the *prpDC* mutant provokes deficiencies in in vitro or ex vivo growth due to the absence of regular B12 supplementation in these experimental conditions. However, the presence of B12 in mice fed with routine diets allows the utilization of the alternative B12 methylmalonate pathway in vivo. Together, these findings highlight the need of in vivo models to study B12-, or others, host-dependent phenotypes.

The emergence of compensatory mutations in the *M. tuberculosis metE* riboswitch under B12 pressure either in vitro, or in the mouse host, emphasizes the importance of B12 for the pathogenesis of the TB bacteria. Inspection of the literature indicates that the *M. tuberculosis* CDC1551 strain, which is naturally defective in the 3'-terminus of *metH*,

and other isogenic *M. tuberculosis metH* mutants, also alleviate the B12-mediated repression through mutations in the *metE* riboswitch when growing under laboratory conditions[23]. Since the CDC1551 is a successful clinical strain which retains a functional B12 riboswitch, we can hypothesize that either this strain has evolved to infect persons with suboptimal B12 levels, or that physiological B12 levels are insufficient to repress the B12 riboswitch. Indeed, we have demonstrated that concentrations as low as 0.01 μg/mL of B12 fails to completely inhibit the *metE* riboswitch, but still allow *metH* functionality in vitro. These results are additional examples that reinforce the key role of the host B12 in the biology of *M. tuberculosis* through regulating the synthesis of an essential aminoacid.

In another context, our present-day society is composed by specific populations affected by deficits in B12[37]. In developing countries, deficiency is much more common, starting in early life and persisting across the life span[38]. In developed countries, B12 deficits affect predominantly to the elderly due to food-bound B12 malabsortion[39], to pregnant women[40], and to vegans and vegetarians improperly supplemented with B12[41]. It is important to remark that veganism/vegetarianism does not necessarily involve a serum B12 deficit. Even if some studies linked the vegetarianism with a higher incidence of TB[42,43], these studies do not report the B12 status of the patients. Accordingly, we cannot exclude B12-independent factors linked to vegetal diets that modulate the TB status in humans.

Our study establishes a direct connection between the optimal B12 serum levels and the virulence of *M. tuberculosis*, which in other words might be translated to lower bacterial virulence in B12 anemic conditions. Since our results come from an animal model, it might be debated that they are challenging to translate into humans. However, we endorse the applicability to humans based on two main arguments. On the one hand, unlike other phenotypes (i.e. immunology, pathogenesis) which are multifactorial, and/or depend on the specific genetic background of the host; the B12 deficiency in our model constitute a single factor (differential B12 serum levels), which closely mimic the human scenario. On the other hand, historic evidence after a *post-mortem* examination of >16.000 autopsies suggests that active TB is profoundly reduced in the context of pernicious anemia[44]. Complementarily, more recent observations indicates a slight increase in the risk of developing TB during the first year of treatment for pernicious anemia[45]. This antagonistic association between anemia and TB has been previously documented for other infectious diseases. A classic example is the protection afforded by sickle cell disease against malaria[46]. Another example is the correlation between higher B12 serum concentration and rising bacterial loads in patients with lepromatous leprosy caused by *M. leprae*[47]. It is key to remember that *M. leprae* is an obligate intracellular pathogen that has experienced a genome downsizing relative to *M. tuberculosis*, and consequently, B12 endogenous production is expected to be abrogated in this pathogen.

Our findings could be also translated into therapeutic approaches to treat TB. In this regard, it has been reported benefits in either prevention or treatment outcomes of TB as an heterologous side effect from the therapy with metformin, a treatment for type 2 diabetes[48]. This and other studies linking lower risk of active TB with metformin treatment have led to propose this drug as an adjunct anti-TB therapy[49]. Despite the accurate mechanism of metformin to ameliorate TB is unknown, a common undesirable effect of metformin is lowering B12 levels in treated patients[50]. According to our study, the B12 decline upon administration of metformin might establish a yet unknown association between this treatment and improvement of TB disease. From our results, it is also appealing to propose new drugable targets in TB bacteria, based on either a chemical blockade of B12 transport, or inhibition of the methionine metabolism in *M. tuberculosis*.

In conclusion, our study, reinterprets the natural evolution of *M. tuberculosis* from its ancestor mycobacteria, shaped by the dependency on a host vitamin. Our findings using the mouse infection model might inform on aspects of human disease under different B12 scenarios, and might contribute to molecularly understand previous historical, clinical, therapeutical and experimental evidences[33,44,45,49].

# Methods

## Ethics statement
This research complies with all relevant ethical regulations. Experimental animal studies were performed in agreement with European and national directives for the protection of animals for experimental purposes. All procedures were carried out under Project Licenses PI 31/21 and PI 43/23, approved by the Ethics Committee for Animal Experiments from the University of Zaragoza.

## Genomic analysis of the B12 biosynthetic pathway
Sequences of each of the 16 *cob* genes present in *M. tuberculosis* H37Rv[10] were aligned using the NCBI Nucleotide BLAST (BLASTn) tool, against the rest of the MTBC genomes available. Mutations were assigned to specific lineages based on inspection of literature and/or description of the sequenced strains. Experimental confirmations of specific mutations were performed in clinical isolates from our laboratory collection previously assigned to MTBC lineages. After DNA extraction from each isolate, the 16 *cob* genes were PCR amplified and sequenced by Sanger methodology to detect polymorphisms.

## Bacterial strains and growth conditions
Mycobacteria were cultured at 37 °C in Middlebrook 7H9 liquid medium (Difco) containing 0.05% (vol/vol) Tween-80 (Sigma) and supplemented with 10% (vol/vol) of ADC (0,5% bovine serum albumin, 0.2% dextrose, 0.085% NaCL and 0.0003% beef catalase) growth supplement (Middlebrook). For solid media, Middlebrook 7H10 broth containing 10% (vol/vol) ADC was used. For B12 uptake in vitro experiments AdoCbl (Sigma-Aldrich) or CNCbl (Sigma-Aldrich) were added to the liquid medium at a concentration of 10 μg/mL. In in vitro growth assays, both liquid and solid were supplemented with AdoCbl and L-Methionine (Sigma-Aldrich) when required at concentrations of 10 μg/mL and 25 μg/mL, respectively. When necessary, 50 μg/mL hygromycin (Hyg) or 20 μg/mL kanamycin (Kan) were added to the media to select mutant strains. Bacterial suspensions of *M. tuberculosis* H37Rv, *M. canettii* C59, and their derivatives used for intranasal infections of mice were properly diluted in PBS from previously quantitated glycerol stock solutions. *Escherichia coli* was cultured in liquid media Luria-Bertani (LB) broth or in solid media LB agar at 37 °C, or at 30 °C for the recombineering event in the case of pKD46 temperature-sensitive plasmid containing strains. When required, media were supplemented with 100 mg/mL ampicillin (Amp), 12.5 mg/mL chloramphenicol (Cm) or 20 mg/mL Kan. A detailed description of each strain used in the study is provided in (Supplementary Table 1).

## Measures of endogenous B12 production and exogenous B12 uptake by bacteria
For B12 production assays, strains from *M. canettii*, MTBC, environmental, and oportunistic mycobacteria were cultured at 37 °C until log phase ($OD_{600} = 0,6$) in 10 ml 7H9-0.05% Tween-80-ADC. Cells were harvested by centrifugation at 4000 x *g* for 10 min and extensively washed three times with sterile PBS 1x. Pellets were resuspended in PBS containing 1% Triton X-100 and transferred into tubes containing glass beads (MP Biomedicals). Suspensions were disrupted by Fast-Prep (6.5 m/s, 45 s) twice and samples were cooled on ice between the cycles. Supernatants containing soluble fractions were filtered through a 0.22 μm-pore-size filter after cold centrifugation at 14000 x g for 5 min. Protein content in the whole-cell extracts were quantified using QuantiPro BCA assay (Sigma Aldrich) for subsequent normalization of

B12 measurements. Bacterial whole-cell extracts were used for the quantification of B12 levels in a Cobas analyser from the hematology section of the University Clinical Hospital "Lozano Blesa" (Zaragoza) based on a competitive electrochemiluminescent binding assay. The values of total B12 obtained in pg/mL were normalized against the total protein values in μg/mL. For B12 uptake assays *M. canettii* and MTBC strains were cultured at 37 °C in 30 ml 7H9-0.05% Tween-80-ADC supplemented when required with AdoCbl (10 μg/mL) and CNCbl (10 μg/mL). Ten mL of culture were taken at day 2, day 7 and 2 months, and bacteria were processed as reported for B12 production studies.

## RNA extraction and quantitative Reverse Transcription-PCR (qRT-PCR)

These protocols were previously described in[51]. Sequences of primers used are specified in (Supplementary Table 2). Melting curves were performed for each amplicon to verify specificity. Four replicates of each gene $C_T$ value were obtained and normalized to the $C_T$ of the *sigA* housekeeping gene (amplified from the same samples), obtaining a $\Delta CT = C_{T,j} - C_{T,sigA}$, where $j$ is a gene different from *sigA*. We calculated a $\Delta\Delta C_T$ value specific for the B21-supplementation condition ($\Delta\Delta C_T(+B12)$), subtracting the $\Delta CT$ mean value for each gene obtained from non-B12-supplemented condition from the $\Delta CT$ obtained for each gene from +B12 condition. Finally, change of expression, denoted as Relative quantity, was calculated with the equation $2^{-\Delta\Delta C_T(+B12)}$.

## RNA-seq analysis

RNA-seq analysis were performed by STAB-vida (Portugal). First, the library construction of cDNA molecules was carried out using a Ribosomal Depletion Library Preparation Kit. The generated DNA fragments (DNA library) were sequenced in the Illumina Novaseq platform, using 150 bp paired-end sequencing reads, and the analysis of the generated raw sequence data was carried out using CLC Genomics Workbench 12.0.3. The bioinformatics analysis started with trimming of raw sequences to generate high quality data only. The high-quality sequencing reads were mapped against the reference genomes, *M. tuberculosis* H37Rv (NC_ 000962.3) or *M. bovis* AF2122/97 (NC_002945.3) when appropriate, using the following parameters: length fraction = 0.8 and similarity fraction = 0.8. The result of mapping served to determine the gene expression levels based on the TPM (Transcripts per Million). To analyse variation and bring out strong patterns in the dataset Principal Component Analysis was performed. Differential expression analysis among samples from each strain and replicate was carried out. Fold changes were calculated from the Generalized Linear Model GLM, which corrects for differences in library size between the samples and the effects of confounding factors. The differentially expressed genes were filtered using the condition "Fold change ≥2 or ≤−2" and the genes that fulfilled that condition were listed (Supplementary Data 1–3).

## Targeted proteomics by MRM/MS

The MRM/MS approach was used for the quantification of specific PrpC, PrpD and MetE peptides in whole-cell extracts of *M. tuberculosis* H37Rv and GC1237. Cultures were grown until logarithmic growth in 35 ml 7H9-0.05% Tween-80-ADC supplemented with or without AdoCbl. Cells were washed three times with sterile PBS 1x and pelleted. Pellets were resuspended in a volume equivalent to 10% of the initial culture volume with PBS containing 1% Triton X-100 and transferred into tubes containing glass beads (MP Biomedicals). Suspensions were disrupted by Fast-Prep (6.5 m/s, 45 s) twice and samples were cooled on ice between the cycles. Supernatants containing soluble proteins were filtered through a 0.22 μm-pore-size filter after cold centrifugation at 14000 x g for 5 min. 1.5 mL of each bacterial extract was taken and a volume equivalent to 10% of the initial volume of TCA. After overnight cold precipitation of the samples and a long cold centrifugation at 14000 x g, the supernatants were discarded, and the

pellets washed with acetone. A second centrifugation was performed for 10 min and the supernatants were discarded again. The pellets were left at room temperature until completely dry. Finally, they were resuspended in 1 mL of 100 mM Tris solution and the total proteins present in the final samples were quantified using QuantiPro BCA assay (Sigma Aldrich). The MRM/MS was performed as previously described[52]. The identified and quantified peptides were as follows: for PrpC, GLAGVVVDTTAISK, GELPTDAELALFSQR and VVPQTNSLTYR; for PrpD, IIDNAAVSAASMVR, FTELADGVVEPVEQQR and VPLPAPGEPK; and for MetE, SWLAFGAEK, VPSAEEMADSLR, IEAIVASGAHR and NVDEVTASLHNMVAAAR.

## Construction of mutant strains in *M. canettii* and *M. tuberculosis*

Deletion of *cobM* and *cobK* genes in *M. canettii* C59 was achieved using the BAC-recombineering (BAC-rec) strategy[53]. Briefly, the thermo-sensitive plasmid pKD46 containing the red recombinase from lambda phage[54] was co-transformed into the *E. coli* DH10B clone carrying BAC Rv412 (containing *cobMK* genes)[55]. DH10B Rv412 pKD46 transformants incubated with arabinose 0.15% were subsequently transformed with a PCR product obtained using "KO BAC cobMK Mcan-P1 Fw" and "KO BAC cobMK Mcan P1-Rv" primers. This PCR product consists on a Kan resistance cassette (Kan^R), with FRT sites from pKD4, flanked by 40 bp identity arms to the genes of interest. Recombinants were selected on LB agar containing Kan incubated at 30 °C overnight, and genes deletion in the BAC was confirmed by PCR amplification using "Conf-KO BAC cobMK Mcan-Fw" and "P1 inv" primers for the 5' end and "P2 inv long" and "Conf-KO BAC cobMK Mcan-Rv" for the 3' end. Allelic exchange substrates (AES) containing the Kan^R flanked by ~1 kb identity arms for site-specific recombination were obtained by high fidelity PCR using BAC Rv412-Δ*cobM,K*::Kan^R as template and "Conf-KO BAC cobMK-Fw"/"Conf-KO BAC cobMK-Rv" primers. AESs sequences are shown in (Supplementary Note 1). AES were transformed by electroporation in *M. canettii* C59 carrying pJV53H recombineering plasmid[56] and cultured in the presence of 0.2% acetamide. The BAC-rec strategy was also used in *M. tuberculosis* H37Rv for construction of a Δ*metH*::Kan^R mutant. We followed the procedure described above but starting from BAC Rv73 containing the gene Rv2124c (*metH*). The AES (Supplementary Note 1) were obtained using the BAC-knockouts as templates for a PCR using primers described in (Supplementary Table 2). Then, *M. tuberculosis* H37Rv carrying the pJV53H was electroporated with the AES and recombinant colonies were selected and confirmed. The *M. tuberculosis* H37Rv Δ*metE*::Kan^R mutant was constructed using a synthetic AES (GenScript) (Supplementary Note 1) introduced in H37Rv-pJV53H by electroporation. To favor the recombination event and recovery of recombinants, the transformation mixtures were incubated overnight in liquid medium without antibiotic and in the presence of AdoCbl, since MetH requires B12 as cofactor. The AESs used for the construction of H37Rv Δ*metE*::Kan^R and Δ*metH*::Kan^R mutants were also electroporated in the *M. canettii* C59 strain containing pJV53H in order to obtain *metE* and *metH* mutants in this strain following the same procedure explained above for *M. tuberculosis*. Finally, to construct the double recombinant *M. tuberculosis* H37Rv Δ*metH*::Kan^R Pr_Ag85a_*metE* (Hyg^R), a pMV361H integrative plasmid containing the *metE* gene regulated by the promoter of *fbpA* (Ag85a) (Supplementary Note 1) was electroporated in H37Rv Δ*metH*::Kan^R.

For the final steps of mutant construction and confirmation, recombinant colonies were selected by plating on 7H10-ADC with the appropriate antibiotic/supplement and confirmed by PCR using specific primers (Supplementary Table 2 and Fig. S15). All plasmids used in this work are listed in (Supplementary Table 3).

## Construction of mutant strains in *M. smegmatis*

To construct a *metE* knockout in *M. smegmatis* mc²155 and to mimic the B12 phenotype of *M. tuberculosis*, B12 synthesis in mc²155 was first

abrogated by deletion of *cobLMK* genes. *M. smegmatis* mc²155 carrying pJV53H[56] were electroporated with AES containing the Kan^R flanked by ~50 bp identity arms for site-specific recombination. AES (Supplementary Note 1) were synthesized by PCR amplification of the Kan^R cassette of pKD4 using primers containing 50 bp identity arms (Supplementary Table 2). Transformants were plated on 7H10-ADC-Kan plates and recombinants were confirmed by PCR using specific primers (Supplementary Table 2 and Fig. S12).

Once constructed the *M. smegmatis* mc²155 Δ*cobLMK* strain, and after confirmation loss of pJV53H, this strain was electroporated with the pRES-FLP-*Mtb* plasmid to resolve the Kan^R resistance cassette. The resolved strain was transformed again with the pJV53H to subsequently generate double mutants. For the construction of the *metE* mutant, the AES -which contained a Kan^R cassette and the GFP gene optimized for its expression in mycobacteria (eGFP) (Supplementary Note 1)- was synthesized by GenScript. Construction was confirmed by PCR (Fig. S15) of transformant colonies grown on 7H10-ADC-Kan-L-Met plates.

### Growth of methionine synthesis mutants in laboratory cultures
The L-methionine synthesis mutants were first grown in 7H9-Tween-ADC liquid medium, both with and without B12 (AdoCbl, 10 μg/mL), to assess their growth in the presence and absence of the vitamin and thus validate their phenotypes. To evaluate growth on solid media, in all cases -with the exception of the Δ*metE* mutant, which only grew in the presence of B12 and, therefore, plating with and without B12 was performed from liquid cultures supplemented with B12- the liquid cultures grown without B12 were plated in solid 7H10-ADC plates supplemented with or without AdoCbl (10 μg/mL) and/or L-Met (25 μg/mL).

### Riboswitch structure prediction
Prediction of the secondary structure of the *metE* B12-riboswitch was based on previous studies[23,25]. Using these studies, we identified invariant residues across ~200 B12-riboswitches, and *M. tuberculosis*-specific residues. We also identified a conserved B12-box.

### Mouse model of B12 deficiency
Female C57BL/6 and SCID mice were fed with a diet similar to control group normal diet (Teklad 2014S, ENVIGO), but with no added B12 and with 5% Pectin (Tekland custom diet TD.170206, ENVIGO). Note that pectin has been shown to bind vitamin B12 in the intestine making it less bioavailable[57]. Moreover, cages were renewed more frequent than usual to avoid coprophagia, as it could be an additional source of B12. After 8 weeks of feeding, concentrations of vitamin B12 were measured in plasma. For this, mice were euthanized, and blood was extracted by cardiac puncture. The blood extractions were incubated at 4 °C for 24 h and after centrifugation for 5 min at 4000 x g, the upper phase containing the serum was extracted. Serum samples were analyzed by electrochemiluminescence at the University Clinical Hospital "Lozano Blesa" to quantify the B12-levels. For B12 supplementation experiments, we used the previously mentioned B12-deficient diet (Diet #1 in the text; Tekland custom diet TD.170206, ENVIGO), and a second independent diet reproducing the composition of the standard 2014S diet which was supplemented with a vitamin mix without B12 (Vitamin A, Vitamin D₃, Vitamin E, Vitamin K₃, Vitamin B₁, Vitamin B₂, Niacin, Vitamin B₆, Pantothenic Acid, Biotin, Folate and Choline) to ensure nutritional adequacy after autoclaving (Diet #2 in the text). Subcutaneous B12 supplementation of SCID mice subsequent to *M. tuberculosis* infection, in the corresponding group (Diet #1), consisted on 2.5 μg CNCbl (Optovite B12, Normon laboratories) per week and per mouse, which translates the treatment prescribed for patients with B12 deficits into the animal model.

### Mouse infection experiments
All mice were kept under controlled conditions and observed for any sign of disease. Mouse experimentation and breeding were done in a SPF-facility at 20–24 °C, 50–70% humidity, and a light-dark cycle of 12 h. Immunocompromised SCID and immunocompetent C57BL/6 female mice were infected with a low dose (≈200 CFUs) of the different *M. tuberculosis* wild type and mutant strains and with a high dose (≈50000 CFUs) of the *M. canettii* C59 wild type and the Δ*cobMK* mutant strain by the intranasal route. The animals were anesthetized by inhalation route with Isofluorane (Isboa Vet) using a vaporizer and intranasal administration was performed with two instillations of 20 μl of the bacterial suspension prepared in PBS. Bacterial suspensions for infection were plated in solid agar medium to confirm the CFUs used for in vivo challenges. For survival experiments, SCID mice were monitored daily for the development of clinical signs of disease and examined in case any abnormality of behavior was observed. Weight of mice was followed during the experiment. The euthanasia endpoint was defined at the point that the loss of weight was >20%. Four weeks postinoculation, the bacterial burden was evaluated in the lungs and spleen of C57BL/6 mice. For this, the organs were aseptically removed and homogenized in 1 ml of H₂O using a GentleMacs dissociator (Miltenyi Biotec). CFUs were determined by plating serial dilutions onto 7H10-ADC plates supplemented with AdoCbl when required.

### Statistics
Mice were randomly distributed in groups of 6 animals per cage prior to start experimental procedures. Results were not blinded for analysis. No statistical method was used to calculate sample size in animal experiments. GraphPrism software was used for statistical analysis. Results from mice survival assays are the pool of two independent experiment (total $n = 12$), and the statistical analysis was performed using Log-rank (Mantel-Cox) test. For enumeration of organ CFUs post-infection, we performed two independent experiments with at least 6 mice/group per experiment, and statistical analysis to detect significant differences in organ CFUs in each mice group was performed using using unpaired t-test, unless indicated otherwise. Statistical tests were considered significant at $p < 0.05$. $p$-values are as follows: ****$0.0001 > p$; ***$0.0001 < p \, 0.001 > p$ ; **$0.01 > p > 0.001$; *$0.05 > p > 0.01$; ns: not significant, $p \geq 0.05$.

### Reporting summary
Further information on research design is available in the Nature Portfolio Reporting Summary linked to this article.

## Data availability
The authors declare that all data supporting the findings of this study are available within the paper and its supplementary information. The datasets generated and analysed during the current study are available in the Gene Expression Omnibus (GEO) server with access number GSE232691. Source data are provided with this paper. In case of additional information, it will be provided upon request to the corresponding author. Source data are provided with this paper.

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

## Acknowledgements

The authors would like to acknowledge Carlos Martín and Nacho Aguiló for the critical reading of the manuscript. We also acknowledge the use of the BSL3 facilities from "Servicio General de Apoyo a la Investigación-SAI" of the University of Zaragoza. The authors thank Irene Orera from the Proteomics Facility of the "Centro de Investigación Biomédica de Aragón" for the MRM-MS analysis. Dr. Joaquín Surra Muñoz from "Escuela Politécnica Superior de Huesca" designed and provided a custom B12-deficient diet. Dr. Sofía Samper from the "Hospital Miguel Servet" provided clinical isolates from specific *M. tuberculosis* lineages. Drs. Luis Callén Sevilla and Luis Palomera Bernal, heads of the hematology section, from the "Hospital Clínico Universitario Lozano Blesa" kindly provided advice and facilities for cobalamin measurements. This work was supported by a grant PID2019-104690RB-I00 funded by MCIN/AEI/ 10.13039/501100011033 to J.G.-A. and by a grant FPU17/02909 funded by the Spanish Ministry of Universities to E.C.-P.

## Author contributions

Conceptualization, E.C.-P. and J.G-A.; methodology, E.C.-P., S.U. and J.G.-A.; experimental research, E.C.-P., S.U., A.P., A.B.G and J.G.-A.; writing–original draft, E.C.-P. and J.G.-A.; writing–review and editing, E.C.-P. and J.G.-A.; figure design, E.C.-P. and J.G.-A.; funding acquisition, J.G.-A.; supervision, J.G.-A.

## Competing interests

The authors declare no competing interests.
