## [Peer Review File · Nature Communications]

Dependency on host vitamin B12 has shaped *Mycobacterium tuberculosis* Complex evolutionREVIEWER COMMENTS

Reviewer #1 (Remarks to the Author):

This is a very interesting study on *M. tuberculosis* and its ability to scavenge B12 during host infections. This study shows for the first time that virulence of *Mtb*, an auxotroph for B12, correlates with B12 plasma levels in mice, but that virulence of *M. canettii* a mycobacterium that is proficient in B12 biosynthesis is not dependent on B12 availability in the host. The authors then identify methionine biosynthesis as a possible reason for virulence attenuation because *Mtb* encodes a B12-dependent (MetH) and -independent (MetE) methionine synthase. Infection experiments using MetH and MetE mutants in anemic vs. normal mice indicate B12 dependent attenuation. Overall, the data presented is intriguing but not rigorous enough to make firm claims.

The main criticisms of the paper are:

1. No complementation of knockout strains was conducted and included in the assays. This is problematic because of potential secondary mutations e.g. in PDIM biosynthesis or compensatory mutations like the ones found in the MetH strain are not uncommon.
2. It is claimed that methionine cannot be taken up by *M. tuberculosis*. This stands in stark contrast to published work PMID 26221021, and it is surprising that this work is completely ignored. Did the authors consider supplying higher Methionine concentrations?
3. For mouse infection experiments in C57BL6 mice only one time point is shown, hence it cannot be determined if the small differences in CFUs measured are indeed B12 dependent or potentially stem from differences in inoculum size. Data with several time points, including a 24h harvest should be shown to prove the point that indeed *Mtb* .
4. All mouse experiments seem to have been done only once. This is concerning, because some of the differences shown (e.g. CFUs in BL6 mice) are very small.

The manuscript is generally well written; however, the intro and discussion have sections that sound more like a review article. The story could be written in a more concise way. Also discussion of what their findings mean for potential therapeutic strategies is missing, but could add value to the story.

Reviewer #2 (Remarks to the Author):

This study set out to discover whether host B12 status could affect progression of disease in a murine model of tuberculosis through interplay with *Mycobacterium tuberculosis*. The significance of this work is that it probes the access of *M. tuberculosis* to a desirable cofactor in what is generally considered to be a nutritionally poor space, and this informs on the metabolic consequences of this interplay on pathogenesis.

The work was carried out thoroughly and with attention to detail. All the results were provided for analysis, along with schema for experimental design. The methodology was clear and sufficiently detailed.

In summary, the authors conducted a survey of current mutations in B12 biosynthetic genes amongst *Mtb* complex bacteria, which indicated ongoing decay in this pathway and demonstrated that members of the *Mtb* complex do not synthesize B12 although closely related ancestral lineages as well as the more distantly related environmental bacteria do. They also demonstrated that B12 uptake mechanisms were active for both cyanocobalamin and adenosylcobalamin in H37Rv, *M. africanum* and *M. bovis*.

In both SCID mice and C56BL/6 mice, a B12 deficient diet led to lower serum levels of B12. SCID mice were used for survival studies and C56BL/6 mice were used to assess bacterial burden at 4 weeks. B12-depleted mice survived longer than B12-replete mice, and the bacterial burden in C56BL6 mice was significantly lower at 4 weeks. In contrast, disease progression was identical regardless of host B12 status with *M. canettii*, which synthesizes its own B12.

This finding was explored at a finer level with two different diets lacking B12, with substantially the same outcomes.

The authors conclude that bacterial growth and pathogenesis was affected by host B12 status. The

key question is whether the bacterium can directly access host B12, or if the loss of B12 affects how the host perceives the bacterium.

For these purposes, *metE* and *metH* mutant strains were constructed and assessed in mice for progression of disease.

A $\Delta metE$ mutant was significantly attenuated in SCID and C57BL/6 mice fed with a conventional diet, and even more so upon infection of anemic mice, which implies firstly that the bacteria could still access B12 from anemic mice to support *metH* function and growth, and secondly that even from B12-replete mice, the B12 available was not enough to support *metH* function. Accordingly, the bacterial burden of the *metH* mutant in mice was also lower and mice infected with *metH* mutant survived longer, implicating transport of B12 and partial repression of the *metE* riboswitch in this impairment.

This was supported by infection studies with a constitutive *metE* expressing strain which was not attenuated.

Comments

Firstly, in S3g, the series +b12 and +B12/+Lmet are supposed to represent different conditions but seem to be duplicates of the same plates.

Otherwise, all the experiments were carried out carefully, all controls were present, and this work represents a substantial finding of general interest. However, I think some of the statements in the introduction are misleading, the conclusions may be overstated, and the limitations of the study have not been made clear as set out below.

lines 64-83: As most animals obtain B12 from the diet, are intestinal pathogens not expected to be able to source B12? Ie Where in the gut is B12 taken up by the host, and where do the bacteria reside?

lines 84-94: The references cited do not show that *Listeria* is sensing host B12 to regulate its virulence. B12 regulation of *Listeria* virulence is not an example of host B12-pathogen cross talk as *Listeria* is fully capable of de novo B12 biosynthetic capability (Vásquez et al, 2022; <https://biolres.biomedcentral.com/articles/10.1186/s40659-022-00376-4>). The insignificance of host B12 levels to pathogenesis to bacteria that possess the ability to manufacture B12 de novo is also shown by experiments conducted in the paper with *M. canettii*.

The word "deleterious" is used to describe the B12 biosynthetic pathway (line 151). Do the authors mean decayed?

Although it is clear from this study (and previously shown by Warner et al 2008) that sufficient external B12 pressure can select for riboswitch mutations in *MetH* mutants, such riboswitch mutants weren't directly identified from bacteria recovered from mouse lungs, and instead were only recovered after a laboratory-mediated stress passage through B12-containing media. The authors should make this clearer. I don't think evolution of a riboswitch mutation under these conditions is surprising either, as this has been previously demonstrated, so perhaps "surprising" (line 324) should be changed to "as expected".

The CDC1551 strain is a successful clinical strain that is compromised for *metH* activity and yet retains a functional B12 riboswitch and B12 transport. The paragraph (lines 392 – 399) is therefore misleading since these experiments are not an example of the key role of host B12 in the evolution of riboswitch mutations as implied.

I am not clear on which arguments surrounding drivers for decay of the B12 biosynthetic pathway in *M. tuberculosis* are being proposed. Ie Are the authors of this paper suggesting that early humans would have had sufficient B12 to promote decay of the B12 pathway in *M. tuberculosis*, (lines 352 -357) or are they suggesting that there were sufficient anemic people that occasional mutations in the pathway facilitated better spread in earlier, more dispersed groups of humans and thus became fixed (lines 400-404 and 425 - 435)?

Contrary to this argument are the studies that find vegetarians are at a higher risk of active TB disease than non-vegetarians (<https://www.ncbi.nlm.nih.gov/pmc/articles/PMC473919/>; <https://jcp.bmj.com/content/jclinpath/41/7/759.full.pdf>) Chanarin et al., found the incidence of tuberculosis in vitamin B12-deficient vegetarians was 133 per 1000, compared to 48 per 1000 in patients on varied diets. They suggested that in anemic patients, the failure to convert

methylmalonic acid to succinic acid resulted in increased concentrations of methylmalonic acid which provided a particularly favorable climate for the multiplication of *M. tuberculosis*. They also found that macrophages were impaired in their ability to kill bacteria, suggesting that host B12 deficiencies altered immune function.

Lines 371 – 378: It is evident that in the mouse strains used in this study, a host B12 effect on disease progression is clear. However, I think it is also premature to extrapolate results from this study to all mice (and from there to all humans, see above also). Notably, Smith et al 2020 showed that disease association with mutations in PPE, *mut* and *bacA* were observed in some mice strains, but not in others. This should also be made clear. Are the mouse strains used in this work associated with the groups assessed by Smith et al.?

I think the study claim (line 447-449) that they have reconciled historical, clinical, therapeutic and experimental evidence is exaggerated and should be worded more carefully. The study has certainly contributed greatly to the understanding of the ability of *M. tuberculosis* to access B12 in these mouse models, which might inform on aspects of human disease.

Reviewer #3 (Remarks to the Author):

Applying the anemic-mice model, the Authors demonstrated that the depletion of B12 in serum affects the ability of *Mtb* to develop tuberculosis. Moreover, the Authors evaluated *Mtb metE* and *metH* mutant phenotype in an anemic mice model. However interesting, data confirm the previously reported studies demonstrating the B12-dependent phenotype of *metE* and *metH* mutants (e.g. Warner et al., 2007) and the effect of B12 transport inactivation (*BacA*-mutant) on chronic mouse infection (Domenech et al., 2009). Therefore, as such, the presented results do not introduce much novelty in the field of the B12 role in *Mtb* pathogenesis.

Also, in certain other parts of the Results, the data repeats the previously published studies. Results-Part 1 "Bacteria causing TB in humans"

Lines 136-147; 155-156: The novelty of this part is very limited since the ability to uptake and inability to produce B12 by *M. tuberculosis* was, directly and indirectly, demonstrated in several previous studies: e.g. Warner et al., 2007; Domenech et al., 2009; Gopinath et al., 2013; Minias et al., 2021 along with the demonstration that *metE* promoter response in *Mtb* to the exogenous B12 in KO-*cobIJ* but not KO-*bacA* mutant (Minias et al., 2021). Moreover, as the Authors stated (Lines 142-143), the production of B12 by various strains of non-tuberculous mycobacteria was also previously demonstrated (Minias et al., 2021). The use of some new strains in the analysis is somehow interesting, however, it does not introduce enough novelty considering the chapter title (Line 119).

Results-Part 2-3 "M. tuberculosis exhibits ..." "B12 supplementation ..."

It's the most interesting part of the manuscript because of the anemic mouse model used by the authors. However interesting, data only confirm the previous, *in vitro* made observation that B12 deficiency affects the growth of *Mtb* strains. It could have been assumed that the same deficiency of B12 *in vivo* affects the survival/infection of *Mtb*. Moreover, it was also reported that *Mtb bacA*-KO-mutant (unable to uptake B12) is attenuated in chronic mice infection (Domenech et al., 2009). So, there is not much new information provided by authors regarding the completed experiments.

Results-Part 4 "The core B12-dependent ..." Lines 217; 228-235; 265-267: The whole B12-dependent transcriptome of the *M. tuberculosis* H37Rv, as well as the high level of B12-dependent downregulation of the *prpR*, *prpCD/metE*, have been previously demonstrated and discussed in detail (Pawelczyk et al., 2021; GEO database (GSE175812)). The Authors do not cite and discuss this paper in any part of the manuscript. Again, the introduction of two more strains in the RNA-Seq analysis does not introduce much novelty to previously published transcriptomic results according to the core B12 response.

Lines 242-254 (Results) describe/discuss previously published data and should be transferred into the Discussion section.

Line 240: Defining a group of B12-downregulated Mtb genes as a „B12 regulon“ is not supported by any additional molecular study (e.g. demonstration of direct B12 binding to the specific promoter regions or interaction between B12 and PrpR regulator).

Results-Part 5 “A M. tuberculosis ...” Lines 272-274; 309-312: The in vitro B12-dependent phenotype of Mtb Δ metE and Δ metH mutants have already been published (Warner et al., 2007) therefore, in Results the Authors should focus on the in vivo studies only. Lines 282-286: L-methionine transport ability in Mtb and M. canettii is an interesting original observation.

Results-Part 5 “Suppressive mutations” The suppressor mutations identified in B12 riboswitch were an interesting observation made by the authors, but also quite expected in the event of inactivation of the metH gene.

In conclusion, the authors performed a lot of well-conducted experiments, however, mostly only confirmed the previous knowledge about the mechanisms of metabolism regulation in the presence of B12. The results obtained should be published in a more specialized journal.

Warner DF, et al. J Bacteriol. 2007 May;189(9):3655-9.
Domenech P, et al. J Bacteriol. 2009 Jan;191(2):477-85.
Gopinath K, et al. Open Biol. 2013 Feb 13;3(2):120175.
Minias A, et al. Sci Rep. 2021 Jun 10;11(1):12267.
Pawelczyk J, et al. Sci Rep. 2021 Jun 11;11(1):12396.

Zaragoza, 21th December 2023

Manuscript reference number: NCOMMS-23-08005-T

Manuscript title: Dependency on the host vitamin B12 has shaped the *Mycobacterium tuberculosis* Complex evolution

Dear Reviewers,

We acknowledge your time and consideration during the revision of our manuscript. After carefully reviewing, answering and addressing the reviewer's comments, we are confident that the revised version of the study has been substantially strengthened.

We have reproduced the B12-dependent in vivo attenuation in mice adding additional time points, and including new mice and M. tuberculosis strains. Further, we have characterized some missing points related to the B12-dependent genes in M. tuberculosis. Lastly, we have, revised the main text, not only to address the reviewer's suggestions, but also to remove some exaggerated sentences. The new version of the manuscript contains 16 new, or revised, figure panels and 3 new supplementary figures. A complete list of the new experiments performed is provided below:

- 1) *Survival experiments of SCID mice fed with control, or B12 deficient, diets after a M. tuberculosis H37Rv infection.*
- 2) *Survival experiments of SCID mice fed with control, or B12 deficient, diets after M. canettii C59 infection.*
- 3) *M. tuberculosis H37Rv infection of C57BL/6 mice fed with control, or B12 deficient, diets to enumerate lung CFUs after 24h, 4 weeks, or 8 weeks post-infection*
- 4) *M. canettii C59 infection of C57BL/6 mice fed with control, or B12 deficient, diets to enumerate lung CFUs after 24h, 4 weeks, or 8 weeks post-infection*
- 5) *Survival experiments of SCID mice fed with control, or B12 deficient, diets after infection with the M. tuberculosis metE mutant.*
- 6) *M. tuberculosis metE mutant infection of C57BL/6 mice fed with control, or B12 deficient, diets to enumerate lung CFUs after 24h or 4 weeks post-infection.*
- 7) *Survival experiments of SCID mice fed with control, or B12 deficient, diets after infection with the M. tuberculosis methH mutant*
- 8) *M. tuberculosis methH mutant infection of C57BL/6 mice fed with control, or B12 deficient, diets to enumerate lung CFUs after 24h or 4 weeks post-infection.*
- 9) *Survival experiments of SCID mice fed with control, or B12 deficient, diets after infection with M. tuberculosis methH mutant-complemented.*
- 10) *M. tuberculosis methH mutant-complemented infection of C57BL/6 mice fed with control, or B12 deficient, diets to enumerate lung CFUs after 24h or 4 weeks post-infection.*
- 11) *Demonstration of the B12-dependent phenotype in vivo using a third, independent, mouse strain. DBA/2 mice fed with control and B12-deficient diets were infected with M. tuberculosis and lung CFUs were enumerated 4 weeks*

after infection.

- 12) *Assays of L-methionine uptake by M. tuberculosis in liquid and solid media, to corroborate previous findings by Berney et al. 2015.*
- 13) *Identification of putative compensatory mutations in the metE riboswitch occurring during an in vivo infection.*
- 14) *Construction of a bacA mutant in M. tuberculosis*
- 15) *Survival experiments of SCID mice fed with control, or B12 deficient, diets after infection with the M. tuberculosis bacA mutant*
- 16) *M. tuberculosis bacA mutant infection of C57BL/6 mice fed with control, or B12 deficient, diets to enumerate lung CFUs after 4 weeks post-infection.*
- 17) *Vitamin B12 uptake experiments using the M. tuberculosis bacA mutant.*
- 18) *Dose-dependent assays using various concentrations of vitamin B12 to identify the in vitro requirements of B12 by Meth.*
- 19) *Characterization of the inhibitory effect of vitamin B12 over the B12 riboswitch by dose-dependent assays.*

Please find enclosed the manuscript text with the revised changes highlighted in yellow, as well as a new version of the figures and supplementary material.

Yours cordially

Jesús Gonzalo-Asensio (on behalf of the co-authors of the manuscript)

REVIEWER COMMENTS

Reviewer #1 (Remarks to the Author):

This is a very interesting study on *M. tuberculosis* and its ability to scavenge B12 during host infections. This study shows for the first time that virulence of *Mtb*, an auxotroph for B12, correlates with B12 plasma levels in mice, but that virulence of *M. canettii* a mycobacterium that is proficient in B12 biosynthesis is not dependent on B12 availability in the host. The authors then identify methionine biosynthesis as a possible reason for virulence attenuation because *Mtb* encodes a B12-dependent (MetH) and -independent (MetE) methionine synthase. Infection experiments using MetH and MetE mutants in anemic vs. normal mice indicate B12 dependent attenuation. Overall, the data presented is intriguing but not rigorous enough to make firm claims.

We acknowledge the overall positive impression of the reviewer praising the novelty of the study. We have conducted additional in vitro and in vivo experiments to improve the rigor of the presented data.

The main criticisms of the paper are:

1. No complementation of knockout strains was conducted and included in the assays. This is problematic because of potential secondary mutations e.g. in PDIM biosynthesis or compensatory mutations like the ones found in the MetH strain are not uncommon.

The reviewer is right, indeed from our previous works, we are confident that compensatory/secondary mutations might arise. However, this time, we decided not to include complemented strains based on two main claims:

- *In other virulence studies, the use of different *M. tuberculosis* strains (i.e. a wild type and a mutant) is the main differential factor to be interrogated. Thus, the emergence of undesired mutation(s) in the mutant strain could be ruled out by introducing complemented strain in the experiment. However, in the present study, we interrogate the phenotype of the same *M. tuberculosis* strain, assayed in the same mice background, being the B12 status of the animals the only differential factor to be interrogated. Accordingly, the assayed *M. tuberculosis* strain acts as a self-control in each experiment. In this same context, we decided to confirm that the differing B12 status of the animal is responsible for differences in *M. tuberculosis* virulence by conducting the B12 supplementation experiment shown in Figure 2. This experiment demonstrated that B12-deficient mice supplemented with B12 showed equivalent phenotype to control mice with normal B12 levels.*
- *By using *metE* and *metH* mutants, we anticipated that both strains should show complementary virulence phenotypes. While the *metE* mutant is more attenuated under B12 deficiency, the *metH* mutant is more attenuated when B12 is present. Showing two complementary phenotypes add robustness to the study by demonstrating the main hypothesis with two independent *M. tuberculosis* strains.*

*Nevertheless, our data support the absence of additional mutations in the *metH* mutant,*

at least in the context of the differential virulence in the variant B12 scenarios:

- On one hand, the behavior of the *metH* mutant under B12 deficiency (Figures 4f, 4g) is indistinguishable from the wild type strain. Differences between the wild type and the *metH* mutant are exclusively observed under physiological B12 levels (Figures 4d, 4e) due to the inhibition of *metE* by its B12 riboswitch.
- The above-mentioned assumption is reinforced by the fact that complementation of the *metH* mutant with a B12-independent *metE* gene show the same phenotypes than the wild type in either control, or B12 deficient, mice (Figure 5).

For the *metE* mutant, we can rule out the emergence of impactful mutations leading to virulence attenuation, such those in PDIM synthesis, based on these observations:

- From our previous works with the live attenuated *M. tuberculosis* MTBVAC vaccine, which is based on a *phoP*+*fadD26* double deletion mutant, we have generated robust data related to the *fadD26* mutant, which is defective in PDIM synthesis. This mutant is highly attenuated in a SCID mice model, showing survival times longer than 90 days relative to the 28-30 days survival of the wild type-infected group. Thus, survival times of the *metE* mutant (35 and 60 days with or without B12, respectively) rules out the presence of impactful mutations as those in PDIM biosynthesis. For organ CFU, we and other, have observed a reduction of at least 3-log in lung CFUs when examining PDIM defective mutants, which contrast with the 1.56- and 0.75-log reduction in the *metE* mutant relative to the wild type under control, or B12-deficit conditions, respectively.
- Based on this previous experience with the MTBVAC vaccine, and the chance to recover spontaneous mutations in virulence lipids, we routinely check our mutants for neutral red staining, a cytochemical staining indicative of defects in the cell envelope composition.
- In answering another comment from this reviewer, we have repeated the experiments in mice, as will be explained below. Since we recovered and stored various clones when generating the *metE* mutant, we decided to repeat the virulence experiments with an independent *metE* mutant. Results with both *metE* isogenic mutants showed equivalent phenotypes. In our opinion, these results reduce the probability to assign the observed attenuation in the *metE* mutant to unrelated mutations.

2. It is claimed that methionine cannot be taken up by *M. tuberculosis*. This stands in stark contrast to published work PMID 26221021, and it is surprising that this work is completely ignored. Did the authors consider supplying higher Methionine concentrations?

The reviewer is right. We sincerely apologize for omitting this relevant reference, and their implications in the present work. We have conducted new experiments to demonstrate L-methionine transport in M. tuberculosis when growing in liquid and solid media.

*First, we aimed at reproducing those experiments reported in Berney et al, who used a Δ *metA* auxotroph mutant able to recover growth in liquid media supplemented with 50*

*μg/mL L-Met. We also expanded the L-Met range to include 25, 50 and 100 μg/mL L-Met concentrations. We observed that the ΔmetE mutant supplemented with the different L-Met concentrations showed equivalent growth in liquid media to the ΔmetE mutant supplemented with B12, which acts as growth control (New Supplementary Figure 13). Similarly, the ΔmetH mutant supplemented with B12 and different L-Met concentrations grew at comparable levels to the ΔmetH mutant without B12, which serves as control (Supplementary Figure 13). Altogether, these observations are in line with those reported in Berney et al.*

*However, when examining the CFU growth on solid media under the above-mentioned conditions, we observed sharp differences relative to growth in liquid media. The ΔmetE mutant was unable to growth even at the highest L-Met concentration (Supplementary Figure 13). Complementarily, the ΔmetH mutant supplemented with L-Met and B12 did not completely recover the growth of the mutant cultured without B12. The observed growth of the ΔmetH mutant supplemented with L-Met and B12 is indistinguishable from its growth under the sole presence of B12 (Supplementary Figure 13), indicative that those colonies are putative mutants in the metE riboswitch, as we will demonstrate in the answer for another reviewer's comment.*

Overall, we hypothesize that either L-Met bioavailability is higher in liquid media than in solid media facilitating its assimilation by *M. tuberculosis*, or that *M. tuberculosis* express L-Met transport mechanisms preferentially under planktonic growth. These phenotypes appear exclusive of *M. tuberculosis*, since as demonstrated in the initial version of the manuscript, *M. smegmatis*, or *M. canettii*, are able to uptake L-Met when growing in solid media.

In answering this reviewer's comment, we provide an explanation to the initially apparent controversy between both studies. The text has been amended to reflect this exclusive dependency on L-Met of the *metE* mutant when growing on agar plates. The following paragraphs have been included to introduce the new experiments according to the previous study of Berney et al:

lines 258-265: "However, a previous study demonstrated the ability of a *M. tuberculosis metA* auxotroph to recover growth in liquid media after supplementation with L-methionine (Berney, 2015). Based on this observation we also confirmed the ability of the *metE* mutant to recover planktonic growth upon supplementation with L-methionine (Figure S13). Overall, this dependency on exogenous L-methionine of the *metE* mutant when growing on solid, but not in liquid media, is intriguing. We can hypothesize that either L-methionine bioavailability is higher in liquid media, facilitating its assimilation by *M. tuberculosis*; or that *M. tuberculosis* expresses L-methionine transport mechanisms exclusively under planktonic growth."

Lines 304-306: As previously demonstrated with the *metE* mutant, we confirmed that assimilation of exogenous L-methionine in the *metH* mutant occurs under planktonic growth, but not during growth on agar plates (Figure S13).

3. For mouse infection experiments in C57BL6 mice only one time point is shown, hence it cannot be determined if the small differences in CFUs measured are indeed B12 dependent or potentially stem from differences in inoculum size. Data with several time points, including a 24h harvest should be shown to prove the point that indeed Mtb.^[L-SEP]

We agree with the reviewer. Indeed, as requested in the following reviewer's comment, we have repeated all the mouse experiments including a 24-hour post-infection enumeration of lung CFUs. See the accompanying figure below. Please note that, for *M. canettii* C59, we previously optimized the bacterial inoculum to match equivalent phenotypes showed by *M. tuberculosis*, which explains the 100-fold increase after 24 hours post-infection with *M. canettii* C59, with respect to *M. tuberculosis* strains.

For *M. tuberculosis* H37Rv and *M. canettii* C59 wild type strains, we have also included a 2-month post-infection point, to gain insight into the B12-dependent phenotypes in the chronic infection phase (see below new figure panel 1g).

In addition to the 24-hour harvest, we also provide below the enumeration of the different glycerol stocks used to infect mice in the second round of experiments.

4. All mouse experiments seem to have been done only once. This is concerning, because some of the differences shown (e.g. CFUs in BL6 mice) are very small.

We agree with the reviewer. As previously explained, we have repeated the following experiments:

- a) *M. tuberculosis H37Rv infection of SCID mice fed with control, or B12 deficient, diets to enumerate survival.*
- b) *M. canettii C59 infection of SCID mice fed with control, or B12 deficient, diets to enumerate survival.*
- c) *M. tuberculosis H37Rv infection of C57BL/6 mice fed with control, or B12 deficient, diets to enumerate lung CFUs after 24h, 4 weeks, or 8 weeks post-infection*
- d) *M. canettii C59 infection of C57BL/6 mice fed with control, or B12 deficient, diets to enumerate lung CFUs after 24h, 4 weeks, or 8 weeks post-infection*
- e) *M. tuberculosis metE mutant infection of SCID mice fed with control, or B12 deficient, diets to enumerate survival.*
- f) *M. tuberculosis metE mutant infection of C57BL/6 mice fed with control, or B12 deficient, diets to enumerate lung CFUs after 24h or 4 weeks post-infection.*
- g) *M. tuberculosis methH mutant infection of SCID mice fed with control, or B12 deficient, diets to enumerate survival.*
- h) *M. tuberculosis methH mutant infection of C57BL/6 mice fed with control, or B12 deficient, diets to enumerate lung CFUs after 24h or 4 weeks post-infection.*
- i) *M. tuberculosis methH mutant-complemented infection of SCID mice fed with control, or B12 deficient, diets to enumerate survival.*
- j) *M. tuberculosis methH mutant-complemented infection of C57BL/6 mice fed with control, or B12 deficient, diets to enumerate lung CFUs after 24h or 4 weeks post-infection.*

We acknowledge this reviewer recommendation, since repetition of animal infections has resulted in reproducible results with higher statistical significances. The text has been modified to reflect results of the new animal experiments. A summary of these results is provided below, and additional information about each independent experiment (Exp. 1 and Exp. 2) can be accessed through the Source data.xlsx file:

Groups a) and b) Pool of Exp. 1 and Exp. 2 independent experiments (n=6 mice/experiment; 12 mice/group in total). New Figure 1f.

Groups c) and d)

Exp. 1 showing 24 h, 2 weeks and 8 weeks post-infection points. New Figure 1g

Exp. 2 showing 2 weeks post-infection (Supplementary)

Group e) Pool of Exp. 1 and Exp. 2 independent experiments (n=6 mice/experiment; 12 mice/group in total). New Figures 3h and 3j

Group f)

Exp. 1 showing 24h and 2 weeks post-infection
New Figures 3h and 3j

Exp. 2 showing 2 weeks post-infection
(Supplementary)

Group g) Pool of Exp. 1 and Exp. 2 independent experiments ($n=6$ mice/experiment; 12 mice/group in total). New Figures 4d and 4f

Group h)

Exp. 1 showing 24h and 2 weeks post-infection
New Figures 4e and 4g

Exp. 2 showing 2 weeks post-infection
(Supplementary)

Group i) Pool of Exp. 1 and Exp. 2 independent experiments (n=6 mice/experiment; 12 mice/group in total). New Figures 5d and 5f

Group j)

Exp. 1 showing 24h and 2 weeks post-infection points. New Figures 5e and 5g

Exp. 2 showing 2 weeks post-infection (Supplementary)

The manuscript is generally well written; however, the intro and discussion have sections that sound more like a review article. The story could be written in a more concise way. Also discussion of what their findings mean for potential therapeutic strategies is missing, but could add value to the story.

We agree with the reviewer. The following paragraphs have been eliminated to provide a more straightforward message to the reader:

- *A plausible explanation for maintaining the B12 biosynthesis pathway in enteric Yersinia and Salmonella is that it is required for utilization of ethanolamine, a metabolite released from enterocytes during inflammation. Thus, the presence of B12-dependent ethanolamine respiration would enable enteropathogenic Enterobacteriaceae to utilize nutrients in the anaerobic environment of the gut and to outcompete other microorganisms from the microbiota. In this latter context, B12 is also necessary for Salmonella utilization of 1,2-propanediol, a catabolite produced by gut microbes fermenting fucose or rhamnose, common constituents of plant cell walls and intestinal epithelial cells lining the gut.*
- *A solid molecular evidence for host-pathogen signalling mediated by B12 is provided by the intracellular pathogen Listeria monocytogenes. In this bacterium, a B12 riboswitch regulates the expression of a noncoding regulatory RNA, which in turns controls the expression of enzymes involved in ethanolamine utilization, that also require B12 as a cofactor. In fact, defects in ethanolamine utilization or in its regulation by the noncoding RNA attenuated Listeria virulence in mice. Another example is the presence of a B12 riboswitch in L. monocytogenes which controls transcription of a noncoding RNA involved in the regulation of the antisense gene pocR. In the presence of B12, this regulatory mechanism allows the PocR transcription factor to activate the expression of genes which mediate propanediol catabolism and are involved in pathogenesis. Together, both mechanisms integrate a way to sense host B12 to regulate the virulence of this intracellular pathogen.*
- *The human-adapted variants include 6 lineages (L) of M. tuberculosis (L1 to L4, L7 and L8), and 3 lineages of M. africanum (L5, L6 and L9), while the animal-adapted variants are grouped into 4 animal clades (A1-A4).*
- *Indeed, molecular B12 sensing by Listeria to modulate virulence has been already demonstrated.*
- *Another line of evidence endorsing the implications of B12 in the host-pathogen cross-talk is the production of the antimicrobial itaconate by activated macrophages. Itaconyl-CoA is a molecule that structurally resembles methylmalonyl-CoA, which is the substrate employed by the M. tuberculosis methylmalonyl-CoA-mutase MutAB. By molecular mimicry, Itaconyl-CoA strongly inhibits mycobacterial MutAB by undergoing an irreversible covalent attachment to the enzyme's B12 cofactor, and consequently reducing the intrabacterial B12 pool. Additionally, Itaconyl-CoA is also able to inhibit the own human methylmalonyl-CoA-mutase, resulting in inactivation of the human-derived B12 pool. Both Itaconyl-CoA-mediated mechanisms, might well serve to reduce the B12 intracellular availability in infected macrophages as a strategy to fight intracellular infections. Conversely, intracellular pathogens as Y. pestis, or M.*

tuberculosis, have counter evolved mechanisms to degrade itaconate and promote pathogenesis. Altogether, these evidences support the evolutionary arm-race of humans and pathogens to ensure access to B12.

- *At this point, we can theorize that adoption of agriculture 10.000 years ago might have alleviated the TB incidence by putatively decreasing B12 serum levels in Neolithic agrarian populations. It might be also possible that high burdens of TB in the past have selected human genetic variants associated with low B12 serum levels. Indeed, current genome wide association studies, has identified polymorphisms in European, Indian and Chinese populations associated with differential B12 serum concentrations, even if their associations with TB have not been yet established.*
- *Even if reducing B12 serum concentrations under suboptimal levels in active TB patients is not an ethical possibility, maintaining B12 in a narrow physiological range could be an alternative strategy to accelerate recovery during the TB treatment.*

Concerning the therapeutic approaches, we initially mentioned the possible correlation between metformin administration and TB treatment, because a common side effect of metformin is related to lowering B12 levels in treated patients. However, the reviewer is right in pointing out that bacteria-related therapeutic approaches as blockading B12 transport, or inhibition of methionine metabolism, are attractive strategies against TB bacteria. We have included the following paragraph:

- *Lines 448-450: From our results, it is also appealing to propose new drugable targets in TB bacteria, based on either a chemical blockade of B12 transport, or inhibition of the methionine metabolism in M. tuberculosis.*

Reviewer #2 (Remarks to the Author):

This study set out to discover whether host B12 status could affect progression of disease in a murine model of tuberculosis through interplay with Mycobacterium tuberculosis. The significance of this work is that it probes the access of M. tuberculosis to a desirable cofactor in what is generally considered to be a nutritionally poor space, and this informs on the metabolic consequences of this interplay on pathogenesis.

The work was carried out thoroughly and with attention to detail. All the results were provided for analysis, along with schema for experimental design. The methodology was clear and sufficiently detailed. ^[SEP]In summary, the authors conducted a survey of current mutations in B12 biosynthetic genes amongst Mtb complex bacteria, which indicated ongoing decay in this pathway and demonstrated that members of the Mtb complex do not synthesize B12 although closely related ancestral lineages as well as the more distantly related environmental bacteria do. They also demonstrated that B12 uptake mechanisms were active for both cyanocobalamin and adenosylcobalamin in H37Rv, M. africanum and M. bovis.

In both SCID mice and C56BL/6 mice, a B12 deficient diet led to lower serum levels of B12. SCID mice were used for survival studies and C56BL/6 mice were used to assess bacterial burden at 4 weeks. B12-depleted mice survived longer than B12-replete mice,

and the bacterial burden in C56BL6 mice was significantly lower at 4 weeks. In contrast, disease progression was identical regardless of host B12 status with *M. cannetti*, which synthesizes its own B12.

This finding was explored at a finer level with two different diets lacking B12, with substantially the same outcomes. The authors conclude that bacterial growth and pathogenesis was affected by host B12 status. The key question is whether the bacterium can directly access host B12, or if the loss of B12 affects how the host perceives the bacterium.

For these purposes, *metE* and *metH* mutant strains were constructed and assessed in mice for progression of disease. ^{[[SEP]]}A Δ *metE* mutant was significantly attenuated in SCID and C57BL/6 mice fed with a conventional diet, and even more so upon infection of anemic mice, which implies firstly that the bacteria could still access B12 from anemic mice to support *metH* function and growth, and secondly that even from B12-replete mice, the B12 available was not enough to support *metH* function. Accordingly, the bacterial burden of the *metH* mutant in mice was also lower and mice infected with *metH* mutant survived longer, implicating transport of B12 and partial repression of the *metE* riboswitch in this impairment. ^{[[SEP]]}This was supported by infection studies with a constitutive *metE* expressing strain which was not attenuated.

Comments

^{[[SEP]]}Firstly, in S3g, the series +b12 and +B12/+Lmet are supposed to represent different conditions but seem to be duplicates of the same plates.

We sincerely apologize for this error, which arose during assembling this figure panel, and we acknowledge the perceptiveness of the reviewer. We have replaced the B12+L-Met panel with the correct picture, and we have carefully reviewed other similar panels to avoid undesired mistakes.

Otherwise, all the experiments were carried out carefully, all controls were present, and this work represents a substantial finding of general interest. However, I think some of the statements in the introduction are misleading, the conclusions may be overstated, and the limitations of the study have not been made clear as set out below.

We acknowledge the encouraging words of the reviewer, who highlighted the interest, the proper experimental design, and the inclusion of all experimental controls, in our work. We have rewritten the main text to avoid misleading/overstatements, as well as to reflect the limitations of the study. We are at your entire disposal to include additional changes in case that some unspotted sentences have not been amended.

lines 64-83: As most animals obtain B12 from the diet, are intestinal pathogens not expected to be able to source B12? I.e. Where in the gut is B12 taken up by the host, and where do the bacteria reside?

*This question was answered in lines 57-59 of the original version of the manuscript: However, even though some colonic bacteria produce *Cbl*, mammals are not able to*

uptake Cbl produced at this location, since the small intestine is the sole site of absorption.

We propose the following rewriting of this sentence to make a clear distinction between the site where bacteria reside, and the site of B12 absorption:

Lines 57-59: However, even though some bacteria of the microbiota residing in the large intestine produce Cbl, mammals are not able to uptake Cbl produced at this location, since the site of B12 absorption is located in the small intestine

lines 84-94: The references cited do not show that *Listeria* is sensing host B12 to regulate its virulence. B12 regulation of *Listeria* virulence is not an example of host B12-pathogen cross talk as *Listeria* is fully capable of de novo B12 biosynthetic capability (Vásquez et al, 2022; <https://biolres.biomedcentral.com/articles/10.1186/s40659-022-00376-4>). The insignificance of host B12 levels to pathogenesis to bacteria that possess the ability to manufacture B12 de novo is also shown by experiments conducted in the paper with *M. canettii*.

*After carefully reading the manuscript of Vasquez et al. we are not completely confident that *Listeria* is able (or not) to produce endogenous B12. In that manuscript, the authors quantified expression of *cbi* genes involved in B12 synthesis. However, a proper expression of these genes does not exclude the presence of non-functional mutations in the coding regions, similarly to the situation in *M. tuberculosis*.*

*Nevertheless, since this is a controversial observation -and another reviewer have recommended to rewrite the introduction in a more concise way- we consider more appropriate to eliminate the following paragraph: A solid molecular evidence for host-pathogen signalling mediated by B12 is provided by the intracellular pathogen *Listeria monocytogenes*. In this bacterium, a B12 riboswitch regulates the expression of a noncoding regulatory RNA, which in turns controls the expression of enzymes involved in ethanolamine utilization, that also require B12 as a cofactor. In fact, defects in ethanolamine utilization or in its regulation by the noncoding RNA attenuated *Listeria* virulence in mice. Another example is the presence of a B12 riboswitch in *L. monocytogenes* which controls transcription of a noncoding RNA involved in the regulation of the antisense gene *pocR*. In the presence of B12, this regulatory mechanism allows the *PocR* transcription factor to activate the expression of genes which mediate propanediol catabolism and are involved in pathogenesis. Together, both mechanisms integrate a way to sense host B12 to regulate the virulence of this intracellular pathogen.*

The word ‘deleterious’ is used to describe the B12 biosynthetic pathway (line 151). Do the authors mean decayed?

The reviewer is right, we have replaced this word

Although it is clear from this study (and previously shown by Warner et al 2008) that sufficient external B12 pressure can select for riboswitch mutations in *MetH* mutants, such riboswitch mutants weren't directly identified from bacteria recovered from mouse

lungs, and instead were only recovered after a laboratory-mediated stress passage through B12-containing media. The authors should make this clearer. I don't think evolution of a riboswitch mutation under these conditions is surprising either, as this has been previously demonstrated, so perhaps "surprising" (line 324) should be changed to "as expected".

We completely agree with the reviewer. Indeed, since we repeated mice experiments in answering reviewer 1, we decided to select colonies of the methH mutant recovered from mouse lungs to interrogate mutations in the metE riboswitch occurring in vivo. We selected 22 colonies of the methH mutant grown in vitro under B12 pressure, and 30 colonies from mouse lungs fed with normal diet. Results demonstrated that 17/22 colonies grown in vitro contained mutations in the riboswitch. In contrast, only one colony recovered from mouse lungs contained mutations in this region. We thank the reviewer for suggesting this experiment which suggest that the B12 pressure to select metE riboswitch mutations is lower in vivo than in vitro.

See below a screenshot showing the annealing of the selected sequences from mouse lungs colonies, indicating the unique metE riboswitch mutation found in vivo:

We have rephrased the paragraph as follows:

Lines 311-323: We also sequenced the metE riboswitch region from 22 M. tuberculosis methH colonies grown on solid media supplemented with B12, and 30 colonies from lungs of mice infected with this strain and fed with normal diet. As expected, we found that 17/22 of the colonies grown on solid media contained mutations in the riboswitch (Figures 4H and S14). Surprisingly, a single colony from the mouse lungs contained mutations in this region, which might indicate that physiological levels of B12 do not impose a selective pressure as high as that observed in vitro. Mapping of the mutations to the predicted structure of the riboswitch demonstrated that these polymorphisms were regularly distributed, and different polymorphisms affecting invariant residues of B12 riboswitches arose independently in independent colonies (Figure 4I). This result suggests that suppressor mutations in the B12 riboswitch could alleviate the B12

repression of the metE gene, and favour the appearance of M. tuberculosis metH escape mutants when B12 is present, preferentially during growth in vitro.

The CDC1551 strain is a successful clinical strain that is compromised for metH activity and yet retains a functional B12 riboswitch and B12 transport. The paragraph (lines 392 – 399) is therefore misleading since these experiments are not an example of the key role of host B12 in the evolution of riboswitch mutations as implied.

We agree with the reviewer; this paragraph is confusing in its actual form. We propose a rephrasing based on:

- *Demonstration that our M. tuberculosis metH mutant (equivalent to the CDC1551 strain) alleviates the host B12 pressure by accumulating mutations in the riboswitch in vivo.*
- *Demonstration that low concentrations of B12 (0.01 µg/mL) fails to completely inhibit the metE riboswitch, but still allow the metH functionality in vitro. This later observation derived from answering a comment from reviewer 3. See below.*

Accordingly, the new paragraph now reads: (lines 387-397): The emergence of compensatory mutations in the M. tuberculosis metE riboswitch under B12 pressure either in vitro, or in the mouse host, emphasizes the importance of B12 for the physiology of the TB bacteria. Inspection of the literature indicates that the M. tuberculosis CDC1551 strain, which is naturally defective in the 3'-terminus of metH, and other isogenic M. tuberculosis metH mutants, also alleviate the B12-mediated repression through mutations in the metE riboswitch when growing under laboratory conditions. Since the CDC1551 is a successful clinical strain which retains a functional B12 riboswitch, we can hypothesize that either this strain has evolved to infect persons with suboptimal B12 levels, or that physiological B12 levels are insufficient to repress the B12 riboswitch. Indeed, we have demonstrated that concentrations as low as 0.01 µg/mL of B12 fails to completely inhibit the metE riboswitch, but still allow metH functionality in vitro

I am not clear on which arguments surrounding drivers for decay of the B12 biosynthetic pathway in M. tuberculosis are being proposed. Are the authors of this paper suggesting that early humans would have had sufficient B12 to promote decay of the B12 pathway in M. tuberculosis, (lines 352 -357) or are they suggesting that there were sufficient anemic people that occasional mutations in the pathway facilitated better spread in earlier, more dispersed groups of humans and thus became fixed (lines 400-404 and 425 - 435)?

We agree with the reviewer. Discussing the drivers for decay of B12 synthesis in Mycobacterium is highly speculative because we can hardly infer the B12 status of ancient populations. In the same context, applying the ecological theory to the heterogeneity of B12 populations is difficult, and not intuitive. We propose to eliminate the following paragraphs:

- *At this point, we can theorize that adoption of agriculture 10,000 years ago might have alleviated the TB incidence by putatively decreasing B12 serum levels in Neolithic agrarian populations. It might be also possible that high burdens of TB*

in the past have selected human genetic variants associated with low B12 serum levels. Indeed, current genome wide association studies, has identified polymorphisms in European, Indian and Chinese populations associated with differential B12 serum concentrations, even if their associations with TB have not been yet established.

- *This argument somehow recapitulates the ecological theory of microorganisms, which predicts that when virulence is positively correlated with transmission, as is the case in TB, access to a larger number of susceptible hosts favours higher virulence and shorter latency periods. Thus, TB infection of people in developing countries with sub-optimal B12 serum levels, would result in lower virulence, and longer latency times, to guarantee the access to new hosts without decimating the susceptible population.*

Regarding the situation with current human populations, there are three different ideas, which in our opinion are not mutually exclusive:

1. *Intracellular pathogens (Mycobacterium, Listeria, Salmonella) contains B12 sensing mechanisms (i.e. riboswitches), which might be used to detect the intracellular environment and consequently to promote virulence. This is explained in lines 347-356.*
2. *Anemic B12 populations are at lower risk of developing TB. This is detailed in lines 400-420.*
3. *Latency could represent a strategy of M. tuberculosis to restrain its virulence under an unfavorable situation (i.e. B12 deficit due to malnutrition and/or other factors). Accordingly, when the host status is healthier (i.e. optimal B12 levels), M. tuberculosis is able to replicate and promote virulence, as demonstrated in the present study. This is explained in lines 428-439.*

We are at the whole disposal of the reviewer to rewrite/clarify these concepts if they remain confusing.

Contrary to this argument are the studies that find vegetarians are at a higher risk of active TB disease than non-vegetarians (<https://www.ncbi.nlm.nih.gov/pmc/articles/PMC473919/>; <https://jcp.bmj.com/content/jclinpath/41/7/759.full.pdf>) Chanarin et al., found the incidence of tuberculosis in vitamin B12-deficient vegetarians was 133 per 1000, compared to 48 per 1000 in patients on varied diets. They suggested that in anemic patients, the failure to convert methylmalonic acid to succinic acid resulted in increased concentrations of methylmalonic acid which provided a particularly favorable climate for the multiplication of M. tuberculosis. They also found that macrophages were impaired in their ability to kill bacteria, suggesting that host B12 deficiencies altered immune function.

While these studies link vegetarianism with a higher incidence of TB, neither of these reports have explored the B12 status of the patients. Accordingly, we propose to cite both manuscripts in the context that the link between veganism/vegetarianism and TB should be examined carefully. The reviewer comment is very acknowledged, since it is

necessary to alert the reader that, in the absence of proper biochemical markers, it is premature to associate veganism/vegetarianism with infectious diseases.

We propose to include the following paragraph:

Lines 404-408 It is important to remark that veganism/vegetarianism does not necessarily involve a serum B12 deficit. Even if some studies linked the vegetarianism with a higher incidence of TB (Chanarin, 1988, Strachan, 1995), these studies do not report the B12 status of the patients. Accordingly, we cannot exclude B12-independent factors linked to vegetal diets that modulate the TB status in humans.

Lines 371 – 378: It is evident that in the mouse strains used in this study, a host B12 effect on disease progression is clear. However, I think it is also premature to extrapolate results from this study to all mice (and from there to all humans, see above also). Notably, Smith et al 2020 showed that disease association with mutations in PPE, mut and bacA were observed in some mice strains, but not in others. This should also be made clear. Are the mouse strains used in this work associated with the groups assessed by Smith et al.?

This question is difficult to answer with precision. The CC (collaborative cross) mouse strains used in Smith et al. are a panel of recombinant-inbred lines generated by randomizing the genetic diversity of existing mouse resources (A/J, C57BL/6J, 129S1Sv/ImJ, NOD/ShiLtJ, NZO/H1LtJ, CAST/EiJ, PWK/PhJ, and WSB/EiJ). These 8 CC F0 lines were intercrossed to make F1s. Accordingly, the F1s strains used in the study of Smith et al. are mosaic population of F0 strains, as can be viewed in:

<http://csbio.unc.edu/CCstatus/index.py?run=CCV>

<http://csbio.unc.edu/CCstatus/CCGenomes/>

One of the purposes of the study of Smith et al. was recapitulating the heterogeneity of TB disease, because current animal models do not reflect the human diversity. For this objective, Smith et al used the F1s CC mosaic mouse strains infected with a library of M. tuberculosis mutants for associating bacterial genetic requirements with host genetics and immunity.

Thus, even if we have used C57BL/6 (which is a F0 strain), the genetic content of the intercross of this strain with another one will result in a mosaic of both genomes. It is possible that F1s CC mouse strains carry different polymorphisms affecting their susceptibility/resistance to M. tuberculosis. In a context of TB susceptibility depending on the B12 status of the mice, it is also possible that the F1s CC strains carry polymorphisms in B12 assimilation, or B12 metabolic genes.

Overall, it is beautiful to observe that in the context of the mouse host diversity, M. tuberculosis mutations in B12-related genes are down-selected in some F1s mouse (CC061, CC024, CC018, CC003, CC005, CC032, CC015, CC004, CC009, CC039), but not in others. In the context of TB disease depending on the B12 serum levels from the host, this would probably resemble the situation in humans since there is a heterogeneity

in B12 serum levels between humans/populations. However, elucidating whether the polymorphisms of the CC061, CC024, CC018, CC003, CC005, CC032, CC015, CC004, CC009, CC039 are related (or not) to some B12 metabolic genes is clearly out of our knowledge.

The reviewer is right in advising that it is premature to extrapolate results from this study to all mice. However, it is important to note that the two mouse strains analyzed (SCID - which derives from the BALB/c background-, and C7BL/6) are genetically independent. Additionally, we performed a preliminary experiment in a third mouse background, DBA/2, which is known for being more sensitive to *M. tuberculosis* infection, and it represents a different background from SCID and C57BL/6. We observed >2-fold reduction in bacterial load when DBA/2 were fed with a B12-deficient diet relative to the controls and results almost reached statistical significance ($p=0.0581$). Overall, even if we agree with the reviewer in her/his assumption, we are also confident that our results have been reproduced in three independent mice genetic backgrounds.

We propose the following rephrase of the paragraph mentioned by the reviewer:

Lines 359-369: In *Mycobacterium*, a recent study used a combination of genetically diverse mice and a battery of *M. tuberculosis* mutants, with the objective of identifying bacterial virulence requirements in the context of a heterogeneous host genetics and immunity. Among their findings, *bacA*, *mutB*, and *PPE2* mutants were negatively selected when infecting some mice strains. Of this pathway, *bacA* is a *M. tuberculosis* B12 transporter, *mutB* encodes one of the constituents of the B12-dependent MutAB methylmalonyl-CoA-mutase, and *PPE2* is regulated by a B12 riboswitch. Even if this experimental design does not resemble our current study, this independent finding

complementarily supports the importance of B12 during infection in murine models of TB. Nevertheless, it might be argued that our results obtained with two genetically independent, SCID and C57BL/6, laboratory mouse models, could not be extrapolated to all mice strains.

I think the study claim (line 447-449) that they have reconciled historical, clinical, therapeutic and experimental evidence is exaggerated and should be worded more carefully. The study has certainly contributed greatly to the understanding of the ability of *M. tuberculosis* to access B12 in these mouse models, which might inform on aspects of human disease.

We agree with the reviewer. This final paragraph has been rewritten as follows:

*In conclusion, our study, reinterprets the natural evolution of *M. tuberculosis* from its ancestor mycobacteria, shaped by the dependency on a host vitamin. Our findings using the mouse infection model might inform on aspects of human disease under different B12 scenarios, and might contribute to molecularly understand previous historical, clinical, therapeutical and experimental evidences*

Reviewer #3 (Remarks to the Author):

Applying the anemic-mice model, the Authors demonstrated that the depletion of B12 in serum affects the ability of *Mtb* to develop tuberculosis. Moreover, the Authors evaluated *Mtb* *metE* and *metH* mutant phenotype in an anemic mice model. However interesting, data confirm the previously reported studies demonstrating the B12-dependent phenotype of *metE* and *metH* mutants (e.g. Warner et al., 2007) and the effect of B12 transport inactivation (*BacA*- mutant) on chronic mouse infection (Domenech et al., 2009). Therefore, as such, the presented results do not introduce much novelty in the field of the B12 role in *Mtb* pathogenesis.

*We acknowledge this critical point of the reviewer. In fact, the mentioned literature, as well as related studies, are cited and discussed elsewhere in the manuscript. Below, we provide arguments about the novelty and added values of our study, which in our opinion represents a big leap in the context of *M. tuberculosis* pathogenesis in vivo:*

- *The manuscript of Warner et al. explores the implications of *metE* and *metH* exclusively in vitro. Here we complement that work by providing in vivo experiments. It is important to remark that obtaining equivalent results between in vitro and in vivo situations is not always obvious. As an example, when answering to reviewer 1, we demonstrate that L-methionine transport in *M. tuberculosis* occurs in liquid media, but not on solid plates. Accordingly, which of these phenotypes could be expected in vivo? In our opinion, demonstration of bacterial phenotypes in animal models is fundamental to understand its implications in virulence. We do not only study the effect of *metE* and *metH* attenuation in vivo, but also established a mouse model of B12 deficit to study these mutants in a context more translatable to the situation in B12 deficient populations. We should be aware that the B12 deficient mouse model require long term experiments (2 months diet+2 months infection+2 months CFU*

- plates/analysis) which again, represent an added value to this story.
- We complement the study of Warner et al. by providing evidence that *M. tuberculosis*, but not environmental (*M. smegmatis*), or ancestor-like (*M. canettii*) mycobacteria are able to uptake methionine when growing on solid media. These experiments required the construction and characterization of 7 different mutants in different mycobacteria, which in our opinion add value to our study. Further, since our *in vivo* experiments with the *metE* and *metH* mutants resembles the situation of these *M. tuberculosis* mutants growing on agar plates, we can also speculate about the inability of *M. tuberculosis* to assimilate methionine *in vivo*, or the insufficient intracellular methionine concentration to support *M. tuberculosis* in the animals. Please find below the new experiments performed to demonstrate these arguments:

Supplementary Figure S13:

a

100 ug/mL L-Met 50 ug/mL L-Met 25 ug/mL L-Met 10 ug/mL Ado-B12 (control)

M. tuberculosis H37Rv $\Delta metE$

b

c

100 ug/mL L-Met 50 ug/mL L-Met 25 ug/mL L-Met No Ado-B12 (control)
 10 ug/mL AdoB12 10 ug/mL AdoB12 10 ug/mL AdoB12

M. tuberculosis H37Rv $\Delta metH$

d

Figures 3h-k:

Figures 4d-g:

- The manuscript of Domenech et al. demonstrates attenuation of a *M. tuberculosis* *bacA* mutant in B6D2/F1 mice measured by longer survival of mice infected with the mutant relative to the wildtype. However, it is intriguing that when infecting this same mouse strain the bacterial burden in lungs and spleen is indistinguishable between the wild type and the mutant. This result is puzzling since as later demonstrated by Gopinath et al. the *bacA* encodes a B12 transporter in *M. tuberculosis*. Honestly, we were confused with these results since according to our data, lack of B12 utilization by *M. tuberculosis* should result in lower bacterial burden in organs. One possible explanation is related to the different mouse genetic background used between both studies.

Accordingly, we aimed to reproduce the experiments of Domenech et al. using our mouse models of B12 deficit. First, we constructed a *M. tuberculosis* *bacA* mutant in H37rv, the same strain used in Domenech et al. Then, this strain was used in experiments of SCID mice survival (a), and enumeration of organ CFUs in C57BL/6 (b). We obtained equivalent results to those reported in Domenech et al. This is, attenuation of the *bacA* mutant when measuring survival times in SCID, but a phenotype equivalent to the wild type when examining organ replication in C57BL/6. Please find results from these new experiments below:

- Neither the studies of Domenech et al, or Gopinath et al. directly demonstrate the ability of BacA to transport B12 in *M. tuberculosis*. These studies use indirect evidences of bacterial growth under the presence of B12. Thus, we performed direct B12 measurements in the newly constructed bacA mutant exposed to extracellular B12 to confirm, or discard, B12 transport in this strain. This experiment was performed at 2, 7 and 60 days incubation with B12, and we used the wild type strain as B12 assimilation control. Surprisingly, we observed that the bacA mutant is able to assimilate B12 when growing in laboratory media in a phenotype indistinguishable from the wild type (see this new result below). Thus, it is tempting to speculate that other mechanisms for B12 transport are present in *M. tuberculosis* H37Rv, at least during planktonic growth.

- Based on the previous observation, we reinforce the novelty of our findings. Being aware that B12 transport by BacA is a controversial result (due to either different B12 detection methods, strains used, cobalamin isoforms, or concentrations tested between studies), our experiments using the B12 deficient mouse models allows to unequivocally demonstrate the implications of B12 serum levels for virulence in vivo. The unique differential factor between mice groups fed with, or without B12 is the level of serum B12, and consequently, this allows to directly evaluate the impact of this vitamin on the virulence of wild type *M. tuberculosis*. Again, testing this phenotype in animal models is mandatory since bacterial properties in vitro are indistinguishable when cultured with or without B12. Further, the inclusion of the B12 producer, *M. canettii*, in our mouse infection

experiments represents novel results and provide an evolutionary focus of the story, which is missing in previous studies.

- Our mouse experiments demonstrate B12-dependent attenuation in the early and chronic phases of infection with wild type *M. tuberculosis*, based on our new experiments. Further, this is observed in both, survival and organ CFUs enumeration. This phenotype contrasts with that of the *M. tuberculosis* *bacA* mutant which fail to exhibit organ CFUs differences. These results prompt us to hypothesize that other mechanisms, aside from the potential B12 transport, are contributing to the virulence of *M. tuberculosis*, which led us to study the implications of methionine metabolism for *M. tuberculosis* virulence, an aspect previously unexplored *in vivo*.

Figures 1f, 1g

Also, in certain other parts of the Results, the data repeats the previously published studies. Results-Part 1 “Bacteria causing TB in humans ...” Lines 136-147; 155-156: The novelty of this part is very limited since the ability to uptake and inability to produce B12 by *M. tuberculosis* was, directly and indirectly, demonstrated in several previous studies: e.g. Warner et al., 2007; Domenech et al., 2009; Gopinath et al., 2013; Minias et al., 2021 along with the demonstration that *metE* promoter response in *Mtb* to the exogenous B12 in KO-*cobI*J but not KO-*bacA* mutant (Minias et al., 2021).

Moreover, as the Authors stated (Lines 142-143), the production of B12 by various strains of non-tuberculous mycobacteria was also previously demonstrated (Minias et al., 2021). The use of some new strains in the analysis is somehow interesting, however, it does not introduce enough novelty considering the chapter title (Line 119).

We agree with the reviewer, and indeed these references are cited elsewhere. Maybe the more direct evidence of B12 production in non-tuberculosis mycobacteria, but not in *M. tuberculosis* is reported by Minias et al. 2021. Indeed, we have rephrased this sentence to clearly reflect those previous findings by Minias et al:

Lines 122-124: This latter result is in agreement with a recent study demonstrating B12

production in various non-tuberculous mycobacteria, but not in M. tuberculosis

However, a previous work from Minias et al. 2018, proposed that purifying selective pressures suggest that B12 biosynthesis is functional in M. tuberculosis. Clearly, both studies reach contrary conclusions, and our objective was to shed light in this topic. In addition, no information about the M. tuberculosis strains used in Minias et al. 2021 is documented, and accordingly, it cannot be ruled out that lack of B12 production in the M. tuberculosis strains used in Minias et al. 2021 is (or not) lineage-specific.

We started this study in 2014, so it is not surprising that our results partly overlap to previous studies. However, neither of the studies cited by the reviewer explore the ability of bacteria of the M. tuberculosis Complex to produce (or not) vitamin B12. In our opinion, our study explicitly demonstrates for the first time that: i) ancestor-like mycobacteria is able to produce B12; ii) M. tuberculosis Complex from different lineages, including the more widespread lineages 2 and 4, or geographically restricted lineages as M. africanum, are unable to produce B12; iii) animal-adapted strains are unable to produce B12; and iv) environmental and opportunistic mycobacteria are able to produce B12, being this latter point the only which partly overlap with the study of Minias et al.

Lastly, unlike other studies, we provide data about the ability of M. tuberculosis, M. africanum, and M. bovis to uptake exogenous B12. This demonstration was performed with various strains covering the whole M. tuberculosis Complex, different B12 isoforms, and at different growth phases. Overall, we report new, robust and reproducible data which would allow the reader to reach clear conclusions about the B12 production and assimilation of B12 in the M. tuberculosis complex and in non-tuberculous mycobacteria.

Results-Part 2-3 “M. tuberculosis exhibits ...” “B12 supplementation ...”^[SEP]It's the most interesting part of the manuscript because of the anemic mouse model used by the authors. However interesting, data only confirm the previous, in vitro made observation that B12 deficiency affects the growth of Mtb strains. It could have been assumed that the same deficiency of B12 in vivo affects the survival/infection of Mtb. Moreover, it was also reported that Mtb bacA-KO-mutant (unable to uptake B12) is attenuated in chronic mice infection (Domenech et al., 2009). So, there is not much new information provided by authors regarding the completed experiments.

We acknowledge the positive words of the reviewer pointing out this interesting part of the study, and the establishment of a B12 anemic mouse model. We consider that in answering previous comments, we have also addressed this comment. To summarize:

- *Translation of in vitro to in vivo phenotypes is not always obvious, and this is especially important for an intracellular bacterium as M. tuberculosis. There are several examples in the literature reporting contrary observations between in vitro and in vivo data, not only in M. tuberculosis, but in other bacteria.*
- *The phenotypes of a M. tuberculosis bacA mutant in vivo do not completely reproduce the situation in a B12 deficient mouse model infected with wild type M. tuberculosis. Even if results from mice survival are similar, the M. tuberculosis bacA mutant fails to show an attenuated phenotype at the level of organ CFUs.*
- *Given the controversial results obtained from the role of BacA as a B12*

transporter, a B12 deficiency model in vivo is essential to demonstrate the role of host B12 bioavailability in M. tuberculosis virulence, since this phenotype could be hardly explored in in vitro, or even in ex vivo, models.

Results-Part 4 “The core B12-dependent ...” Lines 217; 228-235; 265-267: The whole B12-dependent transcriptome of the M. tuberculosis H37Rv, as well as the high level of B12-dependent downregulation of the prpR, prpCD/metE, have been previously demonstrated and discussed in detail (Pawelczyk et al., 2021; GEO database (GSE175812)). The Authors do not cite and discuss this paper in any part of the manuscript. Again, the introduction of two more strains in the RNA-Seq analysis does not introduce much novelty to previously published transcriptomic results according to the core B12 response.

We humbly and sincerely apologize for omitting this relevant reference. We agree with the reviewer in the need to properly cite and discuss previous findings by Pawelczyk et al. and these sentences have been added to the manuscript:

Lines 222-224: In a previous independent study with the H37Rv strain, the authors also identified PrpR, PrpDC, Rv1132 and MetE as B12-regulated genes, providing robustness and cumulative knowledge about the B12 regulatory network in M. tuberculosis (Pawelczyk, 2021).

Line 370: We and other have studied the effect of B12 on the M. tuberculosis metabolism (Pawelczyk, 2021)

Our previous experience with regulatory networks in M. tuberculosis have shown us the importance to reproduce transcriptomic findings in unrelated strains (either clinically relevant or animal-adapted), because some transcriptomic changes do not necessarily reproduce across the M. tuberculosis Complex. Illustrative examples are as follows: i) The PhoPR virulence regulon is different between M. tuberculosis and animal-adapted strains due to a point mutation in PhoR (PMID: 25049399); ii) Secretion of the virulence factor ESAT-6 is different between the H37Rv laboratory strain and other clinically relevant M. tuberculosis strains due to a polymorphism in the PhoPR-regulated gene whiB6 (PMID: 24891105); iii) The constitutive expression of the DosRS dormancy regulon occurs exclusively in sublineage 2 Beijing strains (PMID: 27799329).

Accordingly, including the H37Rv laboratory strain, the Beijing GC1237 clinical strain, and the AF2122/97 animal-adapted strain add value, reproducibility and robustness to the findings. Otherwise, it is surprising that Pawelczyk et al do not identify PPE2-cobQ1 genes among the top B12-dependent genes since the 5' UTR of PPE2 contains a B12 repressing riboswitch.

Lines 242-254 (Results) describe/discuss previously published data and should be transferred into the Discussion section.

The reviewer is right. We acknowledge this comment since it greatly contributes to the readability of the results section. Paragraph has been moved to lines 370-386.

Line 240: Defining a group of B12-downregulated Mtb genes as a „B12 regulon” is not supported by any additional molecular study (e.g. demonstration of direct B12 binding to the specific promoter regions or interaction between B12 and PrpR regulator).

The reviewer is right. Mentions in the text to the “B12 regulon” have been replaced by “B12-dependent transcriptome” to adequately refer to those genes differentially expressed in response to B12.

Focusing on molecular studies of PrpR, which is not the main objective of this manuscript would result in an unfocused, and confusing, message to the readers. Accordingly, we rather focused on some unexplored aspects of B12 over the methionine metabolism in M. tuberculosis. To our knowledge, previous works have not explored the effects of B12 concentrations over metE transcriptional inhibition, or over MetH allosteric activation. We used our metE and metH mutants to assay various B12 concentrations in new in vitro assays. (a) By using the metE mutant, it is possible to study the dose-dependent effect over MethH. We found that concentrations as low as 0.01 µg/mL are able to support MethH functionality. (b) Conversely, by studying the metH mutant, it is possible to identify those B12 concentrations able to inhibit the metE riboswitch. Again, we found that at 0.01 µg/mL B12, the mutant failed to recover wild type growth, indicative of metE inhibition. Thus, we can hypothesize that M. tuberculosis has evolved to detect concentrations of B12 as low as 0.01 µg/mL, which might be probably present in the microenvironment of the phagolysosome. These results are provided as Supplementary Figure S11 and the following sentences have been added to the results section:

Lines 244-245: Raising B12 concentrations to 0.01 µg/mL partially recovered growth of the metE mutant, and 0.1 µg/mL resulted in a complete growth rescue on agar plates (Supplementary Figure S11).

Lines 295-298: Next, we confirmed that concentrations of 0.01 µg/mL B12 were sufficient to partially inhibit the growth of M. tuberculosis metH on solid plates, otherwise indicative of inhibition of the metE riboswitch (Supplementary Figure S11).

Results-Part 5 “A *M. tuberculosis* ...” Lines 272-274; 309-312: The in vitro B12-dependent phenotype of Mtb Δ metE and Δ metH mutants have already been published (Warner et al., 2007) therefore, in Results the Authors should focus on the in vivo studies only. Lines 282-286: L-methionine transport ability in Mtb and *M. canettii* is an interesting original observation.

We partly agree with the reviewer in focusing exclusively on in vivo studies in the results section. The reviewer is right in spotting that in vitro results with metE and methH mutants have been reported elsewhere. However, our aim is to provide the reader with a complete overview of the experimental flow from transcriptomic identification of B12-dependent genes, going to in vitro studies, and finally to in vivo mouse infection assays. Further, as the reviewer highlight, the L-methionine utilization by M. tuberculosis in solid media is novel and interesting; and this result need to be compared side-by-side with the metE and methH phenotypes in vitro. In addition, as will be answered in next comment, we have performed new B12 assimilation experiments in vitro using the metE and methH mutants, which in our opinion nicely complement the previous in vitro studies.

Results-Part 5 “Suppressive mutations” The suppressor mutations identified in B12 riboswitch were an interesting observation made by the authors, but also quite expected in the event of inactivation of the methH gene.

We wholly agree with the reviewer. To answer reviewers 1, 2 and 3, we performed a new mice infection with the M. tuberculosis methH mutant, and selected colonies recovered

from mouse lungs to interrogate mutations in the *metE* riboswitch occurring *in vivo*. We selected 22 colonies of the *metH* mutant grown *in vitro* under B12 pressure, and 30 colonies from mouse lungs fed with normal diet. Results demonstrated that 17/22 colonies grown *in vitro* contained mutations in the riboswitch. In contrast, only one colony recovered from mouse lungs contained mutations in this region. We greatly acknowledge the reviewer for suggesting this experiment which indicate that the B12 pressure to select *metE* riboswitch mutations is lower *in vivo* than *in vitro*; and accordingly, suppressor mutations are preferably selected *in vitro*.

See below a screenshot showing the annealing of the selected sequences from mouse lungs colonies, indicating the only mutation selected *in vivo*:

We have rephrased the paragraph as follows:

Lines 311-323: We also sequenced the *metE* riboswitch region from 22 *M. tuberculosis metH* colonies grown on solid media supplemented with B12, and 30 colonies from lungs of mice infected with this strain and fed with normal diet. As expected, we found that 17/22 of the colonies grown on solid media contained mutations in the riboswitch (Figures 4H and S14). Surprisingly, a single colony from the mouse lungs contained mutations in this region, which might indicate that physiological levels of B12 do not impose a selective pressure as high as that observed *in vitro*. Mapping of the mutations to the predicted structure of the riboswitch demonstrated that these polymorphisms were regularly distributed, and different polymorphisms affecting invariant residues of B12 riboswitches arose independently in independent colonies (Figure 4I). This result suggests that suppressor mutations in the B12 riboswitch could alleviate the B12 repression of the *metE* gene, and favour the appearance of *M. tuberculosis metH* escape mutants when B12 is present, preferentially during growth *in vitro*.

In conclusion, the authors performed a lot of well-conducted experiments, however, mostly only confirmed the previous knowledge about the mechanisms of metabolism regulation in the presence of B12. The results obtained should be published in a more specialized journal.

Warner DF, et al. J Bacteriol. 2007 May;189(9):3655-9.
Domenech P, et al. J Bacteriol. 2009 Jan;191(2):477-85.
Gopinath K, et al. Open Biol. 2013 Feb 13;3(2):120175.
Minias A, et al. Sci Rep. 2021 Jun 10;11(1):12267.
Pawelczyk J, et al. Sci Rep. 2021 Jun 11;11(1):12396.

We acknowledge the perception of the reviewer about the amount of work reported in this study, and the overall quality of the experiments performed. We have provided arguments throughout the reviewer's comments highlighting the novelty and added value of our work. We try to summarize the main arguments below:

- *Previous findings, with the exception of Domenech et al. 2009 (discussed above), exclusively focused on in vitro phenotypes. It is well known between microbiologists that correlation between in vitro and in vivo outcomes do not always occur, and it is possible to find contrasting phenotypes, especially in virulence studies. As a related example, mutants in the B12-regulated genes *prpR*, *prpC*, or *prpD* showed clear in vitro, or even ex vivo, phenotypes (Munoz-Elias, 2006, Masiewicz, 2012, Griffin, 2012, Pawelczyk, 2021), but failed to reproduce these phenotypes in vivo (Munoz-Elias, 2006, Eoh, 2014, Griffin, 2012), as the mutants behaved indistinguishable from their parent strains.*
- *We provide for the first time an overview of the B12-dependent phenotypes in the *M. tuberculosis* Complex, non-tuberculous mycobacteria, and the ancestor-like mycobacteria. Thus, we provide a novel evolutionary and integrative story about the role of B12 in shaping the virulence of TB (and non TB) bacteria.*
- *We establish for the first time in *M. tuberculosis* research a mouse model of B12 deficit, being a pioneer work to molecularly demonstrate the B12 implications for *M. tuberculosis* virulence in vivo.*
- *In this study, starting in 2014, we have used a collection of 15 wild type mycobacterial strains and we have constructed 8 recombinant strains. These bacteria have been used in diverse in vitro and in vivo studies to robustly, and complementarily demonstrate the reported phenotypes. We consider that our study not only provide novelty with respect to previous works, but it serves to valorize, confirm, and complement those findings by providing in vivo and mechanistic studies.*

REVIEWERS' COMMENTS

Reviewer #1 (Remarks to the Author):

The authors have been responsive to most criticisms and suggestions and have done many additional experiments to support their claims. This is a really comprehensive revision of the manuscript and has improved the story considerably. The only criticism remaining is that of complementation. Without complementation of metE and metH mutants in critical experiments, any claims made based on these results are not on solid ground. The argument that complementation is not necessary because the strains are acting as "self-control" does not preclude the possibility that any phenotype seen is due to a secondary mutation and not the one claimed.

Reviewer #3 (Remarks to the Author):

I would like to thank the authors for their detailed response to the review and for discussing the manuscript in terms of new discoveries. I am convinced that the work of Elena Campos-Pardos et al. is valuable, it organizes an important area of knowledge related to Vit B12-dependent regulation of Mtb metabolism and its impact on Mtb virulence.

I agree with the authors that in vivo studies are extremely valuable because only in this way can we verify whether the in vitro observations are actually important for the pathogenesis process. The revised version of the manuscript, with additional experiments on animals, is much more convincing to me, because the authors, I feel, put much more emphasis on in vivo research and point to it as their main achievements.

In this context, experiments confirming previous discoveries, although I agree with the authors that supplemented with additional detailed information (e.g. B12 synthesis by Mtb lineages), provide a full, well-completed picture of the relationship between the uptake/synthesis of Vit. B12, regulation of Met synthesis and virulence.

Undoubtedly, the greatest value of the work is the introduction of a mouse model of B12 deficit in the aspect of Mtb virulence.

Minor points:

1. The title of the first part of the results should be simplified (line99) e.g. Tubercle bacilli depend on the uptake, not synthesis, of vitamin B12.
2. Discussion, the paragraph describing the role of B12 in mycobacterial latency (lines 428-439) is highly speculative and should be removed

Zaragoza, 26th February 2024

Manuscript reference number: NCOMMS-23-08005B

Manuscript title: Dependency on the host vitamin B12 has shaped the *Mycobacterium tuberculosis* Complex evolution

Dear Reviewers,

We acknowledge your positive impressions after the revision of our manuscript. Please find enclosed the manuscript text with the revised changes. Below you will find responses to reviewer's comments.

Yours cordially

Jesús Gonzalo-Asensio (on behalf of the co-authors of the manuscript)

Reviewer #1 (Remarks to the Author):

The authors have been responsive to most criticisms and suggestions and have done many additional experiments to support their claims. This is a really comprehensive revision of the manuscript and has improved the story considerably. The only criticism remaining is that of complementation. Without complementation of metE and metH mutants in critical experiments, any claims made based on these results are not on solid ground. The argument that complementation is not necessary because the strains are acting as "self-control" does not preclude the possibility that any phenotype seen is due to a secondary mutation and not the one claimed.

We acknowledge the previous reviewer's recommendations since we are confident that this revised manuscript has been considerably improved. Undoubtedly, the role of vitamin B12 over M. tuberculosis pathogenesis represents an important area of research with key implications in basic knowledge of mycobacteria, potential therapeutic applications, or even management of infected patients.

The limitations of the study, including the lack of complemented strains in mouse infections are discussed in the following paragraph (lines 354-358):

"Nevertheless, it might be argued that our results obtained with two genetically independent, SCID and C57BL/6, laboratory mouse models, could not be extrapolated to all mice strains. Another possible limitation is related to the lack of testing of M. tuberculosis Δ metE, or Δ metH, complemented strains in vivo to rule out unrelated virulence phenotypes caused by spontaneous mutations arisen during the mutant construction"

Reviewer #3 (Remarks to the Author):

I would like to thank the authors for their detailed response to the review and for discussing the manuscript in terms of new discoveries. I am convinced that the work of Elena Campos-Pardos et al. is valuable, it organizes an important area of knowledge related to Vit B12-dependent regulation of Mtb metabolism and its impact on Mtb virulence.

I agree with the authors that in vivo studies are extremely valuable because only in this way can we verify whether the in vitro observations are actually important for the pathogenesis process. The revised version of the manuscript, with additional experiments on animals, is much more convincing to me, because the authors, I feel, put much more emphasis on in vivo research and point to it as their main achievements.

In this context, experiments confirming previous discoveries, although I agree with the authors that supplemented with additional detailed information (e.g. B12 synthesis by Mtb lineages), provide a full, well-completed picture of the relationship between the uptake/synthesis of Vit. B12, regulation of Met synthesis and virulence.

Undoubtedly, the greatest value of the work is the introduction of a mouse model of B12 deficit in the aspect of Mtb virulence.

We are grateful for this reviewer for her/his positive words. We are confident that the impact of vitamin B12 over M. tuberculosis virulence is an attractive venue for research.

Minor points:

1. The title of the first part of the results should be simplified (line99) e.g. Tubercle bacilli depend on the uptake, not synthesis, of vitamin B12.

We acknowledge this constructive comment since rewriting of this title results more concise and attractive. See changes in line 87.

2. Discussion, the paragraph describing the role of B12 in mycobacterial latency (lines 428-439) is highly speculative and should be removed

We agree with the reviewer; the following paragraph is merely hypothetical and has been removed together with its accompanying reference. See the deleted paragraph between lines 424-425 with respect to the previous version of the manuscript.

“TB infection does not necessarily imply development of active disease, since *M. tuberculosis*-infected individuals can remain asymptomatic between several months to two years⁴⁸, a period known as latent TB. This latent state might represent an evolutionary strategy of *M. tuberculosis*, and other pathogens, to ensure active infection only when the host environment is more favorable. However, the signal(s) detected by *M. tuberculosis* for the transition from latency to a replicative state are not precisely known. In this regard, it is plausible to propose that when *M. tuberculosis* infects a B12 anemic host, the bacterium restrains virulence mechanisms, and the infected person remains in an asymptomatic latent state. This situation could be prolonged until the B12 status of the infected person improves, and at this stage the bacterium is able to replicate, escaping to the immune control of the host and favouring dissemination to new susceptible individuals. This hypothesis could be tested by a B12 surveillance of latently infected patients who finally developed active TB.”